# hnRNPH1 recruits PTBP2 and SRSF3 to modulate alternative splicing in germ cells

Shenglei Feng[1,4], Jinmei Li[1,4], Hui Wen[1], Kuan Liu[1], Yiqian Gui[1], Yujiao Wen[1], Xiaoli Wang[1] & Shuiqiao Yuan [1,2,3✉]

Coordinated regulation of alternative pre-mRNA splicing is essential for germ cell development. However, the underlying molecular mechanism that controls alternative mRNA expression during germ cell development remains elusive. Herein, we show that hnRNPH1 is highly expressed in the reproductive system and recruits the PTBP2 and SRSF3 to modulate the alternative splicing in germ cells. Conditional knockout *Hnrnph1* in spermatogenic cells causes many abnormal splicing events, thus affecting the genes related to meiosis and communication between germ cells and Sertoli cells. This is characterized by asynapsis of chromosomes and impairment of germ-Sertoli communications, which ultimately leads to male sterility. Markedly, *Hnrnph1* germline-specific mutant female mice are also infertile, and *Hnrnph1*-deficient oocytes exhibit a similar defective synapsis and cell-cell junction as seen in *Hnrnph1*-deficient male germ cells. Collectively, our data support a molecular model wherein hnRNPH1 governs a network of alternative splicing events in germ cells via recruitment of PTBP2 and SRSF3.

---

[1] Institute Reproductive Health, Tongji Medical College, Huazhong University of Science and Technology, Wuhan, Hubei 430030, China. [2] Laboratory Animal Center, Huazhong University of Science and Technology, Wuhan, Hubei 430030, China. [3] Shenzhen Huazhong University of Science and Technology Research Institute, Shenzhen, Guangdong 518057, China. [4]These authors contributed equally: Shenglei Feng, Jinmei Li. ✉email: shuiqiaoyuan@hust.edu.cn

In male mammals, germ cells undergo the following processes: self-renewal of spermatogonial stem cells, proliferation and differentiation of spermatogonia, genomic rearrangement of homologous recombination at meiosis, morphological changes from round spermatids to elongated spermatids, and finally the formation of mature spermatozoa[1]. Numerous studies involving high-throughput sequencing have revealed the important role of alternative pre-mRNA splicing in the transcriptome and proteome diversification during spermatogenesis[2]. Most of the genes involved in spermatogenesis are transcribed and processed into multiple isoforms, mainly through alternative splicing that can expand the form and function of the genome with limited gene number, and this is especially important for highly complex organisms like testis[3]. An interesting example is that of the *Spo11* gene, which encodes for two main protein isoforms (SPO11α and β). The two corresponding transcripts differ for exon 2 skipping (α) or inclusion (β). Early meiotic spermatocytes mainly produce SPO11β, whereas SPO11α function becomes crucial in late meiosis[4,5]. Notably, the timing of *Spo11* alternative splicing parallels that of DSB formation during meiosis, with a first wave occurring in autosomal chromosomes in leptotene-to-zygotene spermatocytes and a second one occurring later, which switches to sex chromosomes in late pachytene spermatocytes. Transgenic mice expressing only SPO11β were fully competent in establishing the first DSB wave, but late foci in sex chromosomes executed by SPO11α were suppressed, leading to male sterility due to inefficient XY pairing and recombination[6]. In addition, other genes with alternative splicing to generate different transcripts encoding numerous protein isoforms were identified, such as *C-kit*[7], *Spata8*[8], *Crem*[9], *Acrbp*[10], and *Ybx3*[11]. Nevertheless, not all spliced variants seem to encode functional proteins. For example, the full-length *Sox17* gene highly expressed in spermatogonia encodes a transcription factor with a high mobility group (HMG) box region in its N-terminus, while an alternatively spliced variant *t-Sox17* in pachytene spermatocytes encodes a truncated protein lacking the entire HMG box region due to the skipping exon and is unable to bind DNA to stimulate transcription[12].

Pre-mRNA splicing is mediated by splicing factors, including the large ribonucleoprotein (RNP) complex known as the spliceosome and particular RNA-binding proteins facilitating the regulation of spliceosome assembly and splice site usage[13]. A growing number of novel splicing regulators involved in spermatogenesis continue to be uncovered by using mouse knockout models, such as SAM68, MRG15, PTBP2, RBM5, and BCAS2[14–18]. Notably, PTBP2 could also control a network of genes involved in cell adhesion and polarity and was essential for Sertoli-germ cell communications[19]. However, the underlying mechanisms of how alternative pre-mRNA splicing functions in germ cell development remain poorly understood.

Heterogeneous nuclear ribonucleoproteins (hnRNPs) are polyvalent RNA-binding proteins with crucial roles in multiple aspects of RNA metabolism, including alternative splicing, mRNA decay, the packaging of nascent transcripts, and translational regulation[20,21]. Some members of the hnRNP family were reported to be highly expressed in male germ cells and involved in the process of spermatogenesis[22,23]. In addition, hnRNP G-T, as an efficient spermatogenic cell-specific splicing factor, can cooperate with RBMY to modulate the signaling pathways in the testis[24]. Among the hnRNP family members, hnRNPH1 has attracted increasing public attention based on its essential role in neurological diseases and cancers[25–27]. However, its physiological function and potential mechanism in the reproductive system still remain unknown.

Here, we find that hnRNPH1 is highly expressed in meiotic cells and is required for pre-mRNA alternative splicing and spermatogenesis. Disruption of hnRNPH1 in male germ cells causes abnormal alternative mRNA splicing and affects the meiosis and germ-Sertoli cell communications, which ultimately leads to male sterility. We further show that hnRNPH1 could directly bind to *SPO11* mRNA and recruit the splicing regulators PTBP2 and SRSF3 to cooperatively regulate the alternative splicing of the target genes. Notably, we find that hnRNPH1 is also essential for oogenesis, and depletion of hnRNPH1 in embryonic female germ cells leads to female infertility with a similar defect in meiosis and cell-cell junction as shown in hnRNPH1-deficient male germ cells. Our data demonstrate a critical role of hnRNPH1 in pre-mRNA alternative splicing of both male and female germ cells and fertility.

## Results

**hnRNPH1 is expressed in meiotic cells and localizes to the chromosomes**. To explore whether hnRNPH1 plays a role in the reproductive system, we first characterized its spatiotemporal expression pattern. High *Hnrnph1* mRNA and protein expression levels were observed in both the testis and ovary (Supplementary Fig. S1a, b). Interestingly, both *Hnrnph1* mRNA and protein expression levels were gradually increased and maintained at a high level from P14 testes (Supplementary Fig. S1c, d). To further clarify the expression pattern of hnRNPH1 during spermatogenesis, its mRNA expression profile was reanalyzed using the previously published single-cell RNA-seq datasets[28]. The result showed that *Hnrnph1* mRNA exists in spermatogonia at low levels, but is highly expressed in spermatocytes followed by early-mid spermatids, and almost absent in late spermatids (Supplementary Fig. S1e, f). In addition, we isolated various types of testicular cells and analyzed the *Hnrnph1* mRNA in these cell populations by RT-qPCR. The result is consistent with those of single-cell RNA-seq analyses (Supplementary Fig. S1g). The subcellular localization of hnRNPH1 in the adult testis was then detected by immunofluorescence (IF) using two different anti-hnRNPH1 bodies, and hnRNPH1 was found to be expressed mainly in spermatocytes, round spermatids, and Sertoli cells but was absent in spermatogonia and elongated spermatids (Fig. 1a, b and Supplementary Fig. S2a–c), suggesting that hnRNPH1 might have a role in meiotic processes. We thus examined the detailed localization of hnRNPH1 during meiosis through chromosome spreads to infer the specific stage at which hnRNPH1 functions. Consequently, the expression of hnRNPH1 was weak in leptotene and zygotene stages, began to increase in the pachytene stage, and maintained the highest level from mid-pachytene to diplotene stage, but was not observed in metaphase (Fig. 1b, c). Interestingly, hnRNPH1 was found to be excluded from the XY body formed in the pachytene stage (Fig. 1c). Next, we examined the localization of hnRNPH1 in meiotic chromosomes of oocytes and found a consistent localization pattern with spermatogenic cells (Supplementary Fig. S2d, e). These results suggest a conserved function of hnRNPH1 in both male and female meiosis during germ cell development.

**hnRNPH1 interplays with splicing factors PTBP2 and SRSF3 in testes**. To elucidate the physiological functions of hnRNPH1 in germ cell development, we performed immunoprecipitation-mass spectrometry (IP-MS) using the hnRNPH1 antibody to unbiasedly identify the interactome of hnRNPH1 in wild-type (WT) testes. Consequently, a total of 171 proteins were identified from the hnRNPH1 antibody pull-down products (Fig. 2a and Supplementary Data 1). Gene Ontology (GO) and KEGG enrichment analyses revealed that 17 of those proteins are related to mRNA splicing (Fig. 2b). Interestingly, among the 16 hnRNPH1-interacting proteins screened by the STRING database (Fig. 2c), three key splicing factors, PTBP2[19], SRSF3[29], and

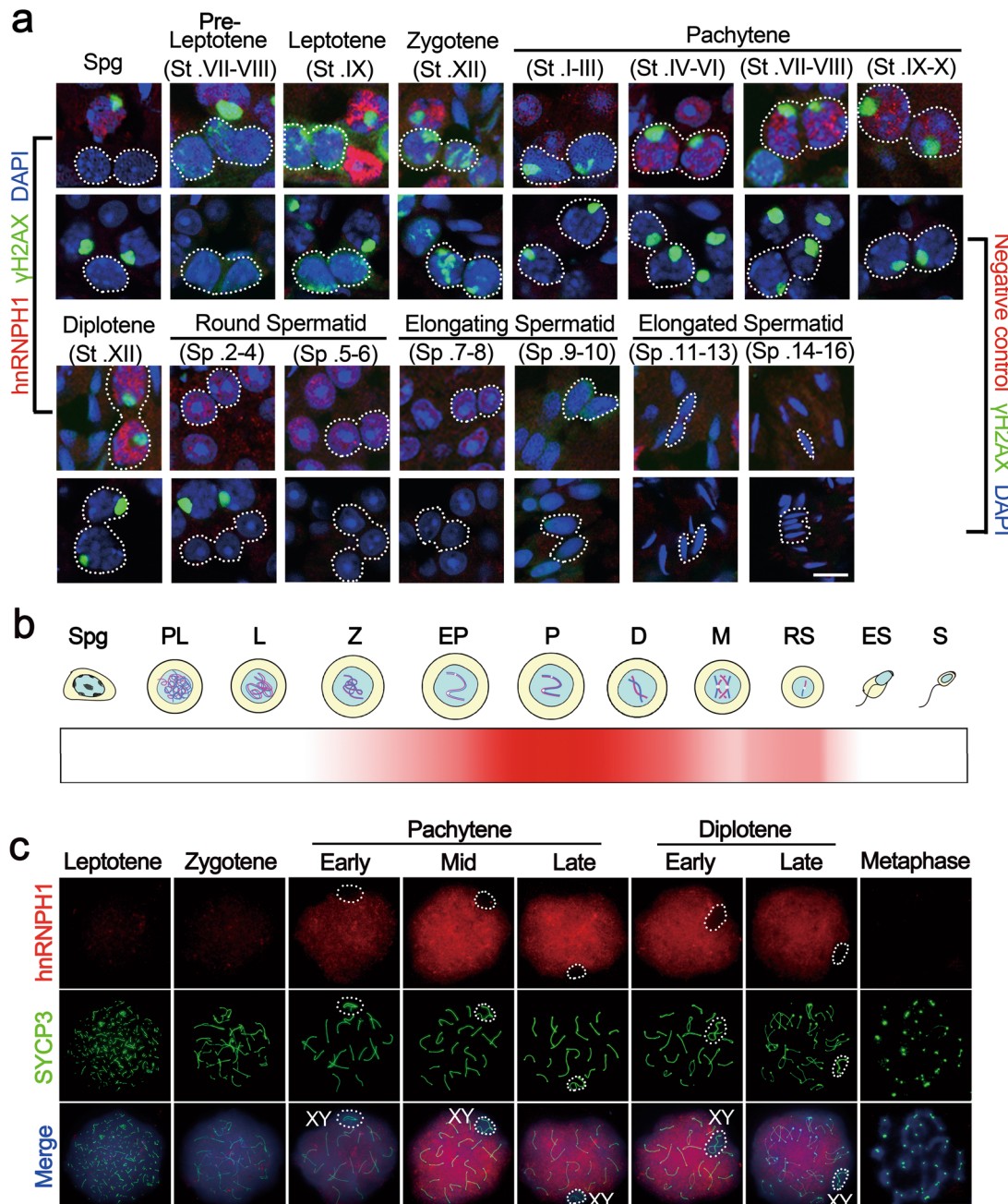

**Fig. 1 hnRNPH1 displays a dynamic expression pattern during spermatogenesis. a** Anti-hnRNPH1 and anti-γH2AX antibodies were used for double immunostaining of wild-type (WT) germ cells from adult testicular cryosections. Lower panels were negative control omitting the primary antibody. The dotted lines outline the indicated cell types. Biologically independent mice (n = 3) were examined in three separate experiments. Scale bars = 10 μm. **b** Time dynamic expression pattern of hnRNPH1 in adult testis during spermatogenesis. Spg spermatogonia, PL pre-leptotene, L leptotene, Z zygotene, EP early pachytene, P pachytene, D diplotene, M metaphase, RS round spermatids, ES elongating spermatids, S spermatozoa. **c** Double immunostaining with hnRNPH1 and SYCP3 on surface-spread spermatocytes from WT adult mice are shown. The dotted lines outline XY body. Biologically independent mice (n = 3) for each genotype were examined. Scale bars = 5 μm.

SNRNP70[14], have been reported to be directly or indirectly involved in spermatogenesis. Through co-immunoprecipitation (Co-IP) assays, we further confirmed that PTBP2 and SRSF3 interacted with hnRNPH1 in the testis (Fig. 2d), while the interaction of SNRNP70 with hnRNPH1 seemed to depend on RNA (Supplementary Fig. S3a). In addition, the IF assay revealed that an obvious co-localization of hnRNPH1 with PTBP2 and SRSF3 was observed in spermatocytes and round spermatids (Fig. 2e), which further supported the interaction effects of hnRNPH1 with PTBP2 and SRSF3 in male germ

cells. More interestingly, highly consistent with hnRNPH1, both PTBP2 and SRSF3 showed an increased expression from the pachytene stage and were excluded from the XY body (Supplementary Fig. S3b).

To further understand the potential mechanism by which hnRNPH1 interacts with PTBP2 and SRSF3 in vitro, we examined their direct association among hnRNPH1, PTBP2, and SRSF3 by ectopically co-expressed hnRNPH1, PTBP2, and SRSF3 in HEK293T cells. Reciprocal Co-IP assays showed that hnRNPH1 could directly bind with PTBP2 and SRSF3 (Fig. 2f–i).

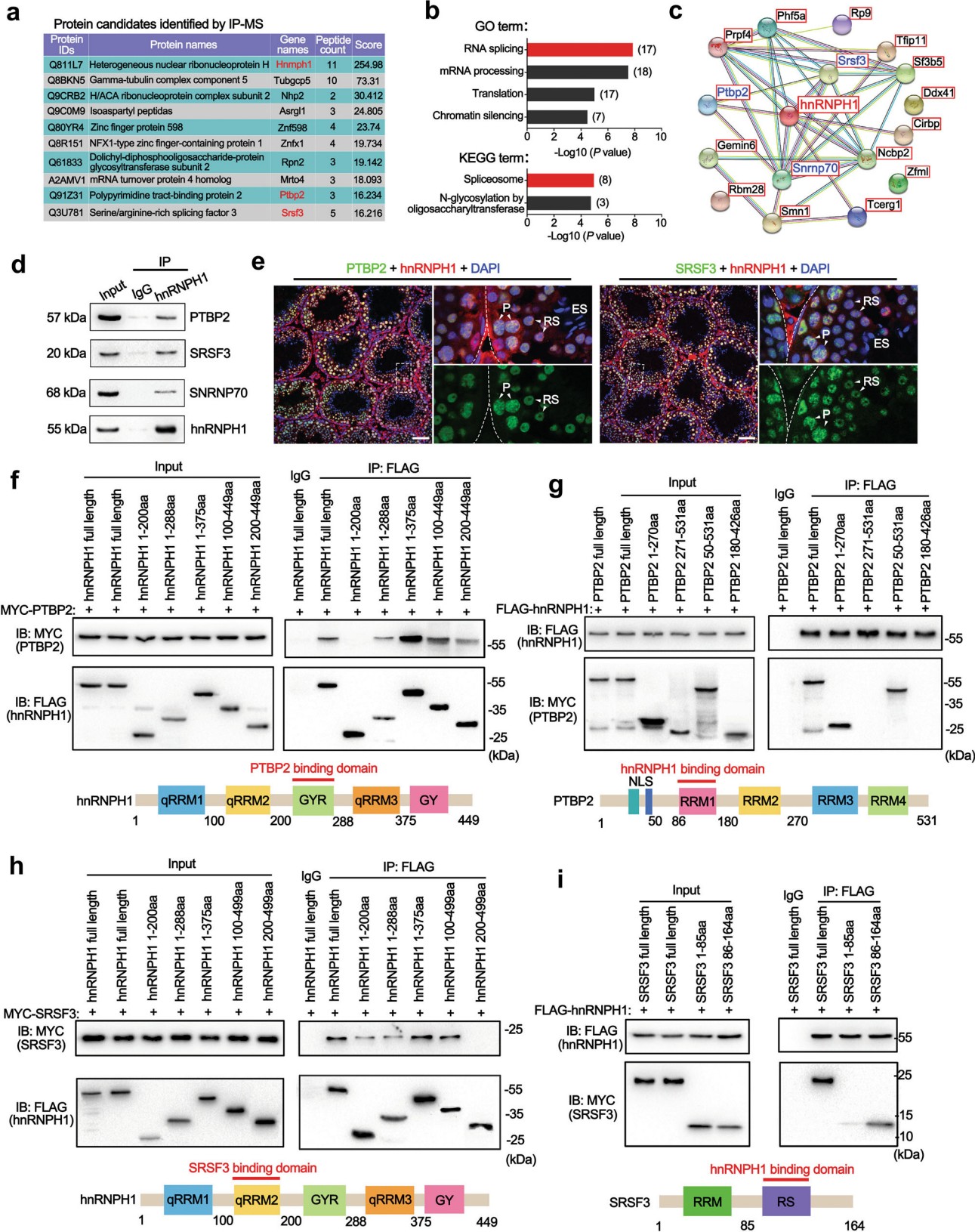

We further found that the 200–288aa region (containing the GRY domain) of hnRNPH1 was necessary for binding to the 50–180aa region (containing the RRM1 domain) of PTBP2 (Fig. 2f, g) and the 100–200aa region (containing the qRRM2 domain) of hnRNPH1 was responsible for interaction with the 86–164aa region (containing the RS domain) of SRSF3 (Fig. 2h, i). These findings suggest that hnRNPH1 cooperates with mRNA splicing factors PTBP2 and SRSF3 through specific regions.

**hnRNPH1 is required for embryonic development and male fertility.** To further determine the physiological roles of hnRNPH1

**Fig. 2 hnRNPH1 interacts with RNA splicing factors PTBP2 and SRSF3. a** A list of ten hnRNPH1-interacting partners in mouse adult WT testes identified by IP-MS is shown. **b** GO and KEGG term enrichment analyses showing the hnRNPH1-interacting proteins identified from IP-MS data. **c** The String 10.5 program (http://string-db.org) analyses showing the protein interaction networks among 17 candidate proteins associated with mRNA splicing. **d** Validation of interactions between hnRNPH1 and three putative hnRNPH1-interacting proteins (PTBP2, SRSF3, and SNRNP70) in mouse testes by in vivo co-immunoprecipitation (Co-IP) assays are shown. IgG was used as a negative control. **e** Co-immunofluorescence staining of PTBP2 (left panels) and SRSF3 (right panels) with hnRNPH1 in adult mouse testes. P pachytene spermatocytes, RS round spermatids, ES elongated spermatids. DNA was stained with DAPI. Scale bars = 50 μm. **f–i** Reciprocal Co-IP assays of interaction domains between hnRNPH1 and its binding partners PTBP2 and SRSF3. HEK293T cells were co-transfected with MYC-PTBP2 and the indicated fragments of FLAG-hnRNPH1(**f**), or co-transfected with FLAG-hnRNPH1 and the indicated fragments of MYC-PTBP2 (**g**), or co-transfected with MYC-SRSF3 and the indicated fragments of FLAG-hnRNPH1(**h**), or co-transfected with FLAG-hnRNPH1 and the indicated fragments of MYC-SRSF3 (**i**), immunoprecipitated with anti-FLAG antibody, and immunoblotted with FLAG and MYC antibodies. Biologically independent experiment (n = 3) were examined (**d–i**). Source data are provided as a source data file.

in the reproductive system, we first generated *Hnrnph1* global knockout mouse line (herein called hnRNPH1 gKO) by crossing germline *Cre* deleter mice with *Hnrnph1*-floxed mice (see the details in materials and methods section) (Supplementary Fig. S4a, b). Surprisingly, we found that hnRNPH1 gKO mice could not develop to full-term birth and died at ~13.5 days of the embryonic stage (Fig. 3a and Supplementary Fig. S4c–e), suggesting that hnRNPH1 is required for mouse embryonic development. These results hinder us from investigating the physiological functions of hnRNPH1 in the reproductive system. We then created germline conditional *Hnrnph1* knockout mice by utilizing the *Stra8-GFPCre* knock-in mouse line[30] to delete exon 6 of the *Hnrnph1* gene in germ cells (Supplementary Fig. S4a, b). *Stra8-GFPCre* induces recombination starting from type A1 spermatogonia in males (before meiosis) and embryonic ovarian germ cells in females (embryonic day 12.5)[30,31]. Western blot (WB), RT-qPCR, and IF analyses showed a significant decrease of both *Hnrnph1* mRNA and protein expression levels in conditional knockout mouse testes (*Hnrnph1;flox/Del Stra8-GFPCre*, hereafter referred to as hnRNPH1 cKO) compared with that of controls (*Hnrnph1flox/flox* or *Hnrnph1;+/flox Stra8-GFPCre*, hereafter called control) and a complete lack of hnRNPH1 protein in all germ cells of hnRNPH1 cKO mouse testes (Fig. 3b–d and Supplementary Fig. 5a), suggesting that germline-specific hnRNPH1 mutants were successfully generated. Interestingly, the hnRNPH1 ablation did not affect the expression of its homolog hnRNPF in testes (Fig. 3b, c) and the localization of its interacting partners PTBP2 and SRSF3 in testes (Fig. 3e, f).

Although hnRNPH1 cKO males were viable and appeared to be normal, they were completely infertile in 5-month-long fertility tests. Testis size from hnRNPH1 cKO males was significantly smaller than their littermate controls (Fig. 3g). The testis/body weight ratio of hnRNPH1 cKO males decreased significantly compared with the control group from postnatal day 28 (P28) onward (Fig. 3h). Histological analyses showed that in adult hnRNPH1 cKO males, there was no detectable mature sperm in seminiferous tubules at stages VII-VIII (Fig. 3i), and spermatogenesis was completely arrested in step 15 spermatids (Supplementary Fig. S5b). In comparison to controls, the apoptotic cells in hnRNPH1 cKO testes were found to be increased significantly by TUNEL assay even at the early P18 stage, suggesting that the development of spermatocytes may have become a defect during the first wave of spermatogenesis (Supplementary Fig. S5c). Interestingly, although round spermatids were histologically normal in hnRNPH1 cKO testes, some of them were prematurely sloughed into epididymis even at P24 (Fig. 3I, j). In addition, no abnormalities were observed in the development of early male germ cells in hnRNPH1 cKO testes at P10 (Supplementary Fig. S5d, e), including undifferentiated spermatogonia (PLZF+) and pre-meiotic germ cells (STRA8+). These results indicate that hnRNPH1 is indispensable for spermatogenesis in mice and

that deletion of hnRNPH1 in spermatogenic cells results in male sterility.

**Ablation of hnRNPH1 causes aberrant mRNA splicing in male germ cells.** Considering that hnRNPH1 is mainly confined to the nucleus of germ cells and interacts with key splicing factors PTBP2 and SRSF3, we speculate that hnRNPH1 is involved in regulating alternative splicing in male germ cells. To test this, we isolated high purity of meiotic pachytene spermatocytes and post-meiotic round spermatids from adult hnRNPH1 cKO and control testes (Supplementary Fig. 6a) and performed high-throughput RNA-seq analysis. As expected, the RNA-seq analysis identified a large number of mRNA splicing-changes in both hnRNPH1 cKO spermatocytes (Supplementary Data 2) and round spermatids (Supplementary Data 3), including skipped exons (SE), alternative 5′ splice sites (A5SS), alternative 3′ splice sites (A3SS), mutually exclusive exons (MXE), and retained introns (RI) (Fig. 4a; |ΔPSI|>10%, P value <0.05), affecting a total of ~800 genes respectively. Approximately 55.8 and 60.1% of AS events were found to be upregulated in hnRNPH1 cKO spermatocytes and round spermatids, respectively (Fig. 4b), suggesting the splicing events repressed by hnRNPH1 were slightly more prevalent than those promoted by hnRNPH1 during spermatogenesis. Interestingly, we found SE was the predominant splicing type in both pachytene spermatocytes (79.6%) and round spermatids (76.3%) among the alternative splicing events affected by hnRNPH1 ablation (Fig. 4c). In addition, hierarchical clustering of differentially expressed genes (DEG) showed that, in pachytene spermatocytes, a total of 3283 genes were upregulated (~91.4%) and 310 genes were downregulated (~8.6%) in hnRNPH1 cKO mice compared to that of control mice. In round spermatids, a total of 2892 genes were upregulated (~82.8%) and 599 genes were downregulated (~17.2%) in hnRNPH1 cKO mice compared to that of control mice (Supplementary Fig. S6b, c and Supplementary Data 4, 5; P value <0.05, fold change >2). These data suggest that hnRNPH1 could affect the transcriptional levels by repressing the gene expression to a large extent during spermatogenesis. Notably, we found that the alternative splicing altered genes in hnRNPH1 cKO pachytene spermatocytes and round spermatids showed an extensive overlap (Fig. 4d). In contrast, there was limited overlap between the differential expression genes and the alternative splicing-changed genes for both pachytene spermatocytes and round spermatids (Supplementary Fig. S6d), suggesting distinct regulation mechanisms of hnRNPH1 in transcription and splicing.

Given that PTBP2 has been reported to regulate the mRNA alternative splicing during spermatogenesis and the ability of hnRNPH1 to interact with PTBP2 (Fig. 2), we asked whether hnRNPH1 could regulate the same alternative splicing genes with PTBP2. To this end, we reanalyzed the published RNA-seq data from *Ptbp2* cKO mice[19] to ensure a consistent analysis criterion with our RNA-seq data analysis of hnRNPH1 cKO mice

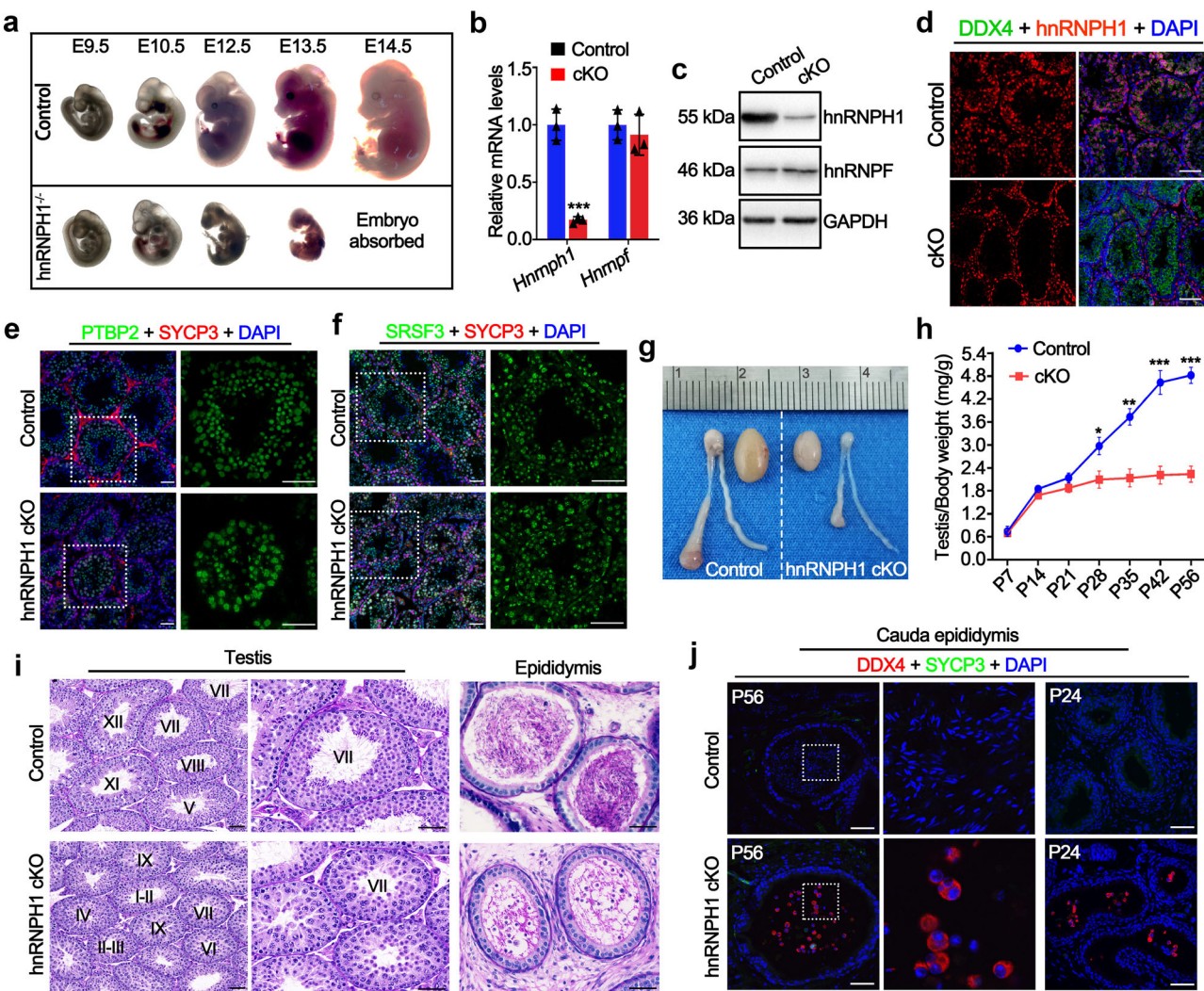

**Fig. 3 hnRNPH1 is indispensable for embryonic development and male fertility in mice. a** Representative embryos at various developmental time points, including embryonic day 9.5 (E9.5), E10.5, E12.5, E13.5, and E14.5, as derived from control and hnRNPH1$^{-/-}$ mice. **b** Real-time qPCR analyses of *Hnrnph1* and *Hnrnpf* mRNA levels in control and hnRNPH1 cKO testes of adult mice. Data were presented as mean ± SD, $n = 3$ (three biological replicates). A two-sided Student's *t*-test was performed, ***$p = 0.0005$. **c** Western blotting analyses of hnRNPH1 and hnRNPF protein in control and hnRNPH1 cKO testes of adult mice. GAPDH was used as a loading control. **d** Immunofluorescence (IF) staining of hnRNPH1 in control and hnRNPH1 cKO testes of adult mice. DDX4 was co-stained to indicate the location of the germ cell. The DNA was stained with DAPI. Scale bars = 50 µm. **e–f** Co-immunofluorescence staining of PTBP2 (**e**) and SRSF3 (**f**) with SYCP3 in testis sections from adult control and hnRNPH1 cKO mice. The DNA was stained with DAPI. Scale bars = 50 µm. **g** Gross morphology of the testes and the epididymides from adult control and hnRNPH1 cKO mice. **h** Testis growth curves of control and hnRNPH1 cKO mice from postnatal day 7 (P7) to P56. Data were presented as mean ± SD, $n = 3$ (three biological replicates). A two-sided Student's *t*-test was performed, left to right: **$p = 0.0089$, ***$p = 0.0009$, ***$p = 0.0004$, ***$p = 0.0001$. **i** Periodic acid-Schiff (PAS) staining of testes and epididymides from control and hnRNPH1 cKO mice at P56. Scale bars = 50 µm. **j** Co-immunofluorescent staining of DDX4 with SYCP3 on control and hnRNPH1 cKO cauda epididymidis sections at P56 and P24, respectively. Scale bars = 50 µm. Biologically independent mice ($n = 3$) were examined in three separate experiments (**c–f**, **i**, **j**). Source data are provided as a source data file.

(Supplementary Data 6). Of note, in PTBP2 null testes, SE also accounted for the most significant proportion (83.2%) among 5 alternative splicing types (Supplementary Fig. S6e). Further analysis revealed that hnRNPH1 and PTBP2 share many alternative splicing genes (Fig. 4d) and events (Fig. 4e) in pachytene spermatocytes and round spermatids. Interestingly, GO term analysis of their shared alternative splicing event-regulated genes highlighted the specific enrichment in functional categories involved in germ cell development, such as "synapsis" and "spermatogenesis" in pachytene spermatocytes (Fig. 4f), as well as "fertilization" and "adhesion junction" in round spermatids (Fig. 4g). Despite poor correlations ($R = −0.024$ in pachytene spermatocytes and $R = 0.023$ in round spermatids) of

$\Delta$PSI values of shared alternative splicing event-regulated genes between hnRNPH1 cKO spermatogenic cells (both spermatocytes and spermatids) and *Ptbp2* cKO testes (Fig. 4h, i), most of the genes involved in germ cell development were regulated in the same direction. Only several of those genes showed abnormal expression levels in hnRNPH1 cKO spermatogenic cells (Supplementary Fig. S6f, g), particularly in spermatids. Taken together, these analyses indicate that hnRNPH1 and PTBP2 could co-regulate the mRNA splicing of some essential genes related to spermatogenesis.

Since the transition from male meiotic spermatocytes to post-meiotic round spermatids is accompanied by genome-wide reprogramming of splicing[2], we next wanted to ask whether the

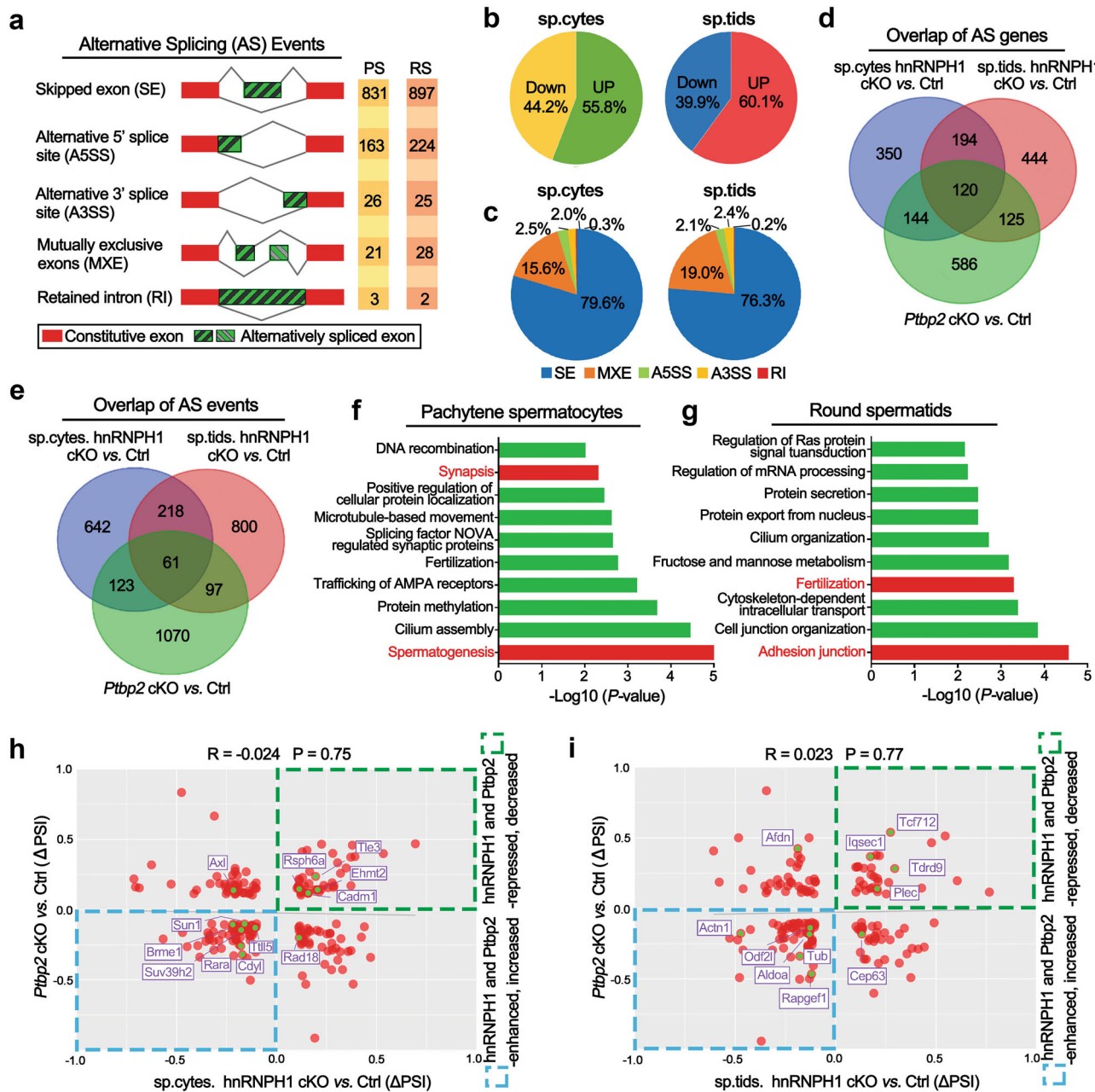

**Fig. 4 hnRNPH1 and PTBP2 synergistically regulate pre-mRNA splicing in spermatogenic cells. a** Summary of differential splicing analysis performed using pachytene spermatocytes (PS) and round spermatids (RS) isolated from the control and hnRNPH1 cKO testes at P25. The numbers of predicted alternative splicing (AS) events in each category upon hnRNPH1 deletion are indicated. **b** Pie charts showing percentages of changed AS events identified in hnRNPH1 cKO versus control spermatocytes (left) and spermatids (right). **c** Pie charts representing the distribution of regulated splicing events among different splicing patterns in hnRNPH1 cKO versus control spermatocytes (left) and spermatids (right). **d** Venn diagrams showing overlap of abnormal AS genes (808) in hnRNPH1 cKO spermatocytes, abnormal AS genes (883) in hnRNPH1 cKO spermatids, and abnormal AS genes (975) in *Ptbp2* cKO testis. **e** Venn diagrams show an overlap of abnormal AS events (1044) in hnRNPH1 cKO spermatocytes, abnormal AS events (1176) in hnRNPH1 cKO spermatids, and abnormal AS events (1351) in *Ptbp2* cKO testis. **f, g** GO term enrichment analyses of genes with the same abnormal AS events caused by hnRNPH1- and PTBP2-ablation in spermatocytes (**f**) and spermatids (**g**). The top ten biological processes ranked by $\log_{10}$ (p value) are listed. **h, i** Distribution of ΔPSI values for genes with abnormal AS events in hnRNPH1 cKO spermatocytes (**h**) and spermatids (**i**). The genes involved in indicated signaling pathways and co-regulated by hnRNPH1 and PTBP2 are marked in red.

genome-wide reprogramming of splicing was affected upon hnRNPH1 deletion during this transition process. By analyzing RNA-seq data, we found that a number of alternative splicing genes or events related to developmental regulation during meiotic spermatocytes to post-meiotic round spermatids transition overlapped with those susceptible to hnRNPH1 ablation in pachytene spermatocytes and round spermatids (Supplementary

Fig. S7a), and the ΔPSI values showed a highly positive correlation ($R = 0.68$, $P = 2.2^{e-16}$) (Supplementary Fig. S7b). Remarkably, we found that both the protein and mRNA levels of hnRNPH1 progressively declined from pachytene spermatocytes to round spermatids during spermatogenesis (Supplementary Fig. S7c, d). Together, these data suggest that hnRNPH1 may be involved in the genome-wide reprogramming of splicing during

meiotic to post-meiotic transition, and its change in expression may contribute to the timely regulated selection of specific alternative splicing events across meiosis.

Of note, many RNA-binding proteins can bind DNA to regulate gene transcription, and we thus reanalyzed the previously ChIP-seq data[32] of hnRNPH1 in HepG2 cells to determine if hnRNPH1 binds DNA to regulate gene transcription. It was found that hnRNPH1 has significant binding peaks near the promoter and the transcription start site (TSS) of more than 10,000 genes (Supplementary Fig. 8a and Supplementary Data 7), suggesting that hnRNPH1 can bind to the promoter of many genes. Combined with our RNA-seq analysis, we found that more than half of the upregulated genes in both spermatocytes and round spermatids overlapped with the genes whose promoters are bound by hnRNPH1 (Supplementary Fig. 8b, c), suggesting that hnRNPH1 may directly bind to the promoters of these genes in germ cells. To test this, ten upregulated genes with the highest fold-changes among these overlapping genes were selected for qPCR verification and it was found that the mRNA expression of 9 genes was significantly increased in hnRNPH1 cKO germ cells (Supplementary Fig. 8d). We then performed ChIP-qPCR using isolated spermatogenic cells and found that the promoters of 4 of the tested genes can be bound by hnRNPH1 (Supplementary Fig. 8e), further suggesting that hnRNPH1 may bind the promoters of these genes in germ cells and regulate their transcription.

**Lack of hnRNPH1 in male germ cells disrupts meiotic processes.** Because some alternative splicing genes related to meiosis, such as synapsis and DNA recombination, were affected in hnRNPH1 cKO spermatocytes (Fig. 4f), we determined whether the splicing levels of corresponding exons of those genes were altered in isolated pachytene spermatocytes. RT-PCR analysis of ten selected alternative splicing genes most closely related to meiosis yielded a 60% validation rate (6/10) (Fig. 5a, b and Supplementary Fig. S9a), indicating relative reliability of the RNA-seq data. WB results further revealed a significant change in protein expression levels of some meiosis-related genes (e.g., *Smc2*, *Cpeb1*, *Ehmt2*, and *Brme1*) in hnRNPH1 cKO spermatocytes compared to controls (Fig. 5c). This suggested that hnRNPH1 ablation may affect the meiosis process by regulating some meiosis-related gene expression at transcriptional and translational levels.

To evaluate the abnormality of hnRNPH1 cKO spermatocytes during meiosis, we checked the meiotic processes by chromosome spreading analyses. In control pachytene spermatocytes, with DSB repair completed, γH2AX signals were removed from autosomes but confined to the sex chromosomes. However, a higher proportion of hnRNPH1 cKO pachytene (29.6% in cKO vs. 3.8% in control) and diplotene (20.5% in cKO vs. 4.6% in control) spermatocytes displayed an abnormal γ-H2A.X distribution compared with controls (Fig. 5d, e), indicating that DNA damage response remained active in synapsed homologs of hnRNPH1-null spermatocytes. Moreover, developmental retardation was observed in hnRNPH1 cKO spermatocyte characterized by a lower proportion of pachytene and diplotene spermatocytes than that of controls (Fig. 5f). At the pachytene stage, each pair of homologous autosomes complete synapsis, while the pairing of sex chromosomes in males only occurs in a small region, known as the pseudoautosomal region (PAR). SYCP1, an important component of the synaptic complex, was used as a marker for synapsis. Interestingly, SYCP1 was present at the PAR between XY chromosomes in control pachynema, while three types of abnormal asynapsis were observed in hnRNPH1 cKO pachynema, including autosomal synapsis (9.2% in cKO vs. 3.6% in

control), mislocalization of SYCP1 to unsynapsed sex chromosomes (14.0% in cKO vs. 2.9% in control), and separated X-Y chromosomes (34.6% in cKO vs. 3.3% in control) (Fig. 5g, h). In addition, although hnRNPH1 deletion did not affect the total number of MLH1 foci (a crossover marker) in pachytene spermatocytes, nearly 30% of the pachytene spermatocytes did not form crossovers on sex chromosomes due to the separation of X-Y (Supplementary Fig. S9b, c). These results indicate that hnRNPH1 ablation caused chromosome asynapsis, especially the high level of unpaired sex chromosomes.

Given previous studies reported that unpaired sex chromosomes in male mice could be caused by abnormal *Spo11* pre-mRNA splicing[6] and hnRNPH1 was identified as a key regulator of *Spo11α* splicing in mouse spermatocytes[33], we next wanted to examine whether the *Spo11* pre-mRNA splicing is changed in hnRNPH1 cKO spermatocytes. Although changed alternative splicing of *Spo11* was not identified in our RNA-Seq data, considering the imperfect accuracy of RNA-seq, we decided to re-examine *Spo11* splicing in P14 and P18 testes, isolated spermatocytes and spermatids from adult testis by RT-PCR, respectively. As previously reported[33], *Spo11* demonstrated significantly changed alternative splicing (*Spo11α/β*) from P14 to P18, and the inclusion of its exon 2 was suppressed at a latter stage (Fig. 5I, j). However, upon hnRNPH1 ablation, this inhibition was relieved, and the expression of exon 2 of *Spo11* maintained a high level at P18 (Fig. 5I, j). Interestingly, this change also existed in isolated spermatocytes but not round spermatids. In addition, abnormal splicing of *Spo11* has been reported to directly cause unsynapsis of sex chromosomes; however, the IF results showed that hnRNPH1 is specifically excluded from the XY body (Fig. 1c), which suggests that hnRNPH1 is unlikely to regulate the synapsis of sex chromosomes directly. Therefore, these results further indicate that hnRNPH1 could indeed regulate the splicing of *Spo11* in spermatocytes, and hnRNPH1-deficiency in male germ cells leads to that alternative splicing of *Spo11* was compromised in the late meiosis, which may be one of the causes of sex-chromosome asynapsis.

**hnRNPH1 deletion impairs the germ-Sertoli cells' communications.** Because a large number of prematurely sloughed round spermatids were observed in hnRNPH1 cKO epididymis (Fig. 3j), it was speculated that the attachment and communication between germ cells and Sertoli cells were destroyed before spermiation at stage VIII of spermatogenesis. To test this, we first detected the RNA splicing of several cell adhesion- and junction-related genes identified from the RNA-Seq data of round spermatids by RT-PCR. Consequently, 8 out of 14 tested genes showed significant RNA alternative splicing-changes in hnRNPH1 cKO round spermatids (Fig. 6a). WB result further confirmed that most alternative splicing-changed genes were also abnormally expressed in protein levels in hnRNPH1 cKO testes (Fig. 6b). Of note, the expression of TCF7L1/2 (two members of TCF family proteins) and IQSEC1 were increased in hnRNPH1 cKO testes, especially in round spermatids (Fig. 6b–d). Based on literature reports, all of these RNA alternative splicing-changed genes were important regulators for Wnt/β-catenin activity[34,35]. While the specific role of these genes in spermatogenesis has not been elucidated, it should be noted that Wnt/β-catenin was a critical cellular signal pathway that is crucial for multiple developmental stages during spermatogenesis[36]. More importantly, we found that β-catenin, known as a component of the cell-cell adhesion apparatus that has been demonstrated to regulate male germ cell differentiation[37,38], displayed an abnormal distribution in hnRNPH1 cKO testes compared with control

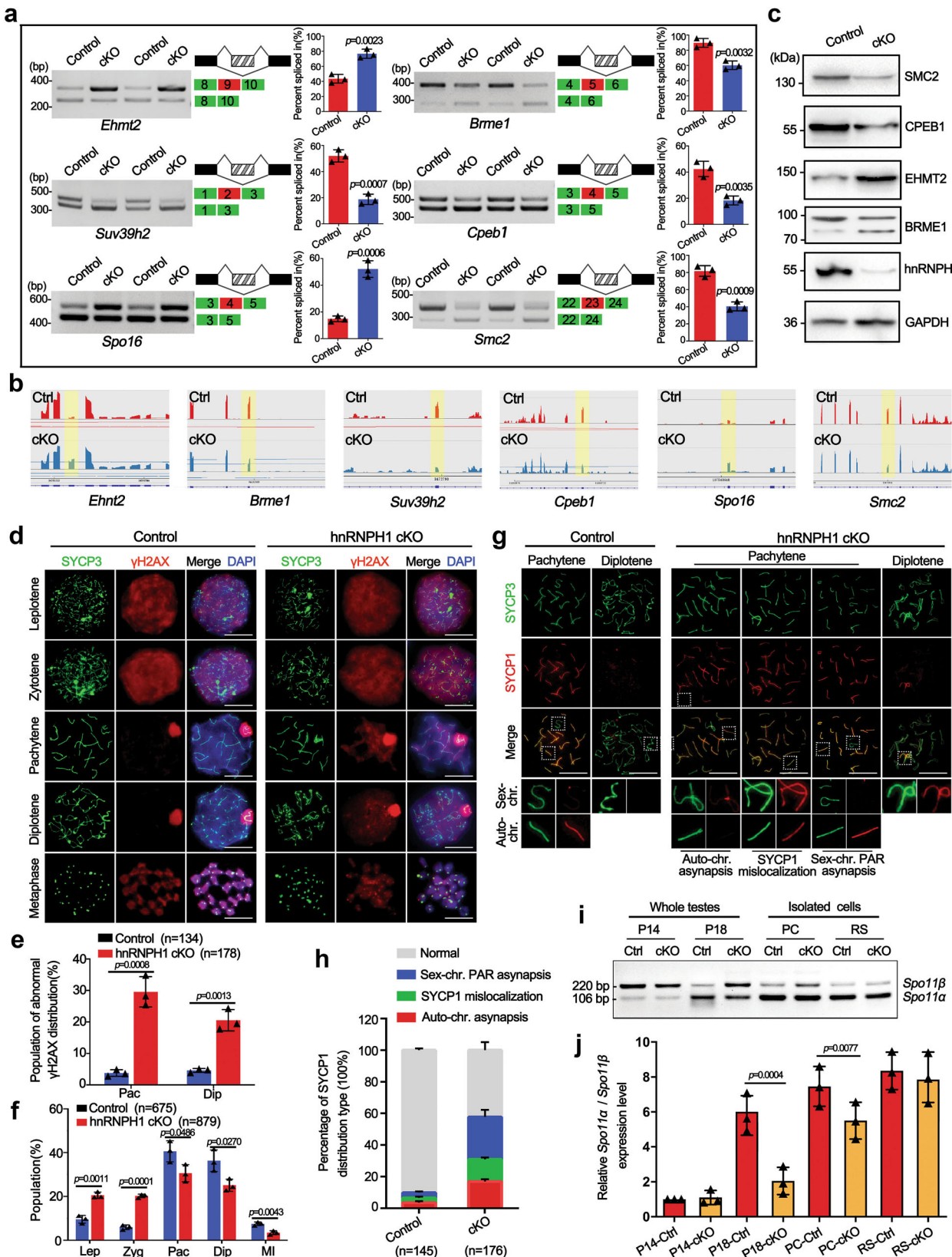

testes in which β-catenin was mainly localized at the blood–testis barrier (BTB) (Fig. 6e).

In addition, given that F-ACTIN regulatory factors were thought to have critical roles in maintaining the adhesion of germ cells with Sertoli cells during germ cell development and movement towards the lumen[39], we next tested the distribution of

F-ACTIN in the hnRNPH1 cKO testes. In control testes, F-ACTIN usually appears in the region where Sertoli cells connect with spermatocytes or elongated spermatids (Fig. 6f). However, whether in adult (P56) or P24 hnRNPH1 cKO testes, F-ACTIN distributed in seminiferous tubules disorderly and abnormally located at the junction between Sertoli cells and

**Fig. 5 hnRNPH1 deficiency in male germ cells causes aberrant AS of meiotic-related genes and defective meiosis. a** RT-PCR analyses for indicated genes between control and hnRNPH1 cKO pachytene spermatocytes. The middle panels represent the schematic diagram of indicated AS exons. Right panels show the quantification of percent spliced in (PSI); Data are presented as mean ± SD, n = 3. **b** RNA-seq results of alternative sites in indicated genes using IGV software analysis. **c** Western blotting analyses of the expression of indicated proteins in control and cKO spermatocytes. GAPDH was used as a loading control. **d** Co-staining of SYCP3 with γH2AX in spermatocyte spreads from control and cKO male mice. Scale bar, 5 μm. **e** Percentages of abnormal γH2AX distribution at the pachytene (Pac) and diplotene (Dip) stages in the control and cKO mice. Data were presented as mean ± SD. A total of n = 134 control and n = 178 cKO spreads were counted from three biologically independent mice for each genotype. **f** Percentages of spermatocytes at the leptotene (Lep), zygotene (Zyg), pachytene (Pac), diplotene (Dip), and metaphase (MI) stages in the control and cKO mice. A total of n = 675 control and n = 879 cKO spreads were counted from three biologically independent mice for each genotype. **g** Co-immunofluorescence staining of SYCP3 with SYCP1 in spermatocyte spreads at pachytene and diplotene stages from control and cKO male mice. Scale bars, 5 μm. **h** Percentages of SYCP1 distribution types in pachytene spermatocytes from the control and cKO mice. A total of n = 145 control and n = 176 cKO spermatocytes were counted from three biologically independent mice for each genotype. **i** RT-PCR analyses of *Spo11β* and *Spo11α* expression levels in control and cKO whole testes at the P14 and P18, as well as spermatocytes and round spermatids isolated from control and cKO mice. **j** Relative *Spo11α / Spo11β* mRNA expression level in indicated groups are shown. Data are presented as mean ± SD, n = 3. Statistical tests: two-sided Student's *t*-test (**a**, **e**, **f**, **h**, **j**). Biologically independent mice (n = 3) were examined in three separate experiments (**c**, **d**, **g**). Source data are provided as a source data file.

round spermatids (Fig. 6f), which is akin to the phenotype observed in *Ptbp2* cKO testes[19]. Moreover, among the genes identified with alternative splicing and protein level changes, *Rapgef6* and *Fndc3a* have been reported to mediate Sertoli cell-spermatid adhesion during mouse spermatogenesis, and both *Rapgef6* and *Fndc3a* knockout mice showed male sterility[40,41]. Together, these results indicate that abnormal alternative splicing of the genes related to cell adhesion in hnRNPH1 cKO spermatids leads to impairment of communication and junction between Sertoli cells and germ cells, thereby causing the phenotype of prematurely sloughed round spermatids in hnRNPH1 cKO testes.

**hnRNPH1 is essential for oogenesis and female fertility.** Considering that the *Stra8-GFPCre* used in this study can effectively induce deletion of the target gene in both male and female germlines, we thus aimed to define the physiological function of hnRNPH1 in oogenesis and female fertility. After 5 months of fertility testing, all hnRNPH1 cKO female mice were also showing complete sterility. We further performed HE stainings on the adult and P21 ovaries and found that the size of hnRNPH1 cKO ovaries was significantly smaller than controls, and almost no follicles were observed in hnRNPH1 cKO ovaries (Supplementary Fig. S10a). These results suggest a crucial role of hnRNPH1 in oogenesis and folliculogenesis. Interestingly, PTBP2 and SRSF3 were also found to be highly expressed in early oocytes, although their expression and localization were not affected in hnRNPH1 cKO ovaries (Supplementary Fig. S10b, c).

To explore the molecular reasons for infertility in hnRNPH1 cKO females, we examined the expression of hnRNPH1 in the ovaries at embryonic day 17.5 (E17.5), when most female germ cells have developed to the pachytene stage. IF results showed that hnRNPH1 was strongly expressed in oocytes and granulosa cells of control mice but specifically absent in the oocytes of hnRNPH1 cKO mice (Fig. 7a), indicating that hnRNPH1 was successfully and specifically knocked out in oocytes. Since hnRNPH1 was localized in the meiotic stage of early oocytes and exhibited a similar expression pattern with spermatocytes (Supplementary Fig. S2d, e), the oocyte development efficiency in the ovaries at P1 and P3 by immunochemistry using a DDX4 antibody (a germ cell marker) to determine if hnRNPH1 functions in the meiosis of early oocytes. Compared with the control, the number of oocytes in the hnRNPH1 cKO ovary at P1 was significantly reduced, and very few oocytes existed at the P3 ovary (Fig. 7b). In addition, we performed immunoprecipitation for hnRNPH1 and found that, as in testes, hnRNPH1 could interact with PTBP2 and SRSF3 in E17.5 ovaries (Supplementary Fig. S10d). Moreover, the TUNEL assay further revealed a substantially higher proportion of apoptotic oocytes in the hnRNPH1 cKO ovary at P0

(Supplementary Fig. S10e). These results indicate that hnRNPH1 is essential for the early development of oocytes and plays a similar function in oocytes as male germ cells.

To further determine the sub-stage of meiotic defects that occur in hnRNPH1 cKO oocytes, we first examined the meiotic processes of oocytes by chromosome spreading analyses at E17.5 ovaries. Similar to hnRNPH1 cKO spermatocytes, an abnormal γH2AX signal appeared in some hnRNPH1 cKO oocytes at the pachytene stage (Fig. 7c). Moreover, a higher percentage of significant asynapsis was observed in hnRNPH1 cKO oocytes compared to control oocytes based on the co-localization of SYCP1 and SYCP3 (Fig. 7d). Given the similar spatiotemporal expression and defective meiosis phenotype between spermatocytes and oocytes, we next asked whether hnRNPH1 regulates meiosis in oocytes by regulating RNA alternative splicing as in spermatocytes. As expected, two genes were identified, *Brme1* and *Cpeb1*, that have been reported to be involved in oogenesis and spermatogenesis[42–44], and which exhibited abnormal alternative splicing in hnRNPH1 cKO ovaries (Fig. 7e). Some cell junction-related genes with aberrant alternative splicing in hnRNPH1 cKO testes, such as *Tcf7l1*, *Tcf7l2*, and *Tcf3*, were abnormally spliced in hnRNPH1 cKO ovaries as well (Fig. 7e). Likewise, TCF7L1/2 and their regulated protein β-catenin were also significantly increased in hnRNPH1 cKO oocytes (Fig. 7f–h). In addition, since hnRNPH1 can directly bind to *Tcf3* and control its splicing to regulate the expression of E-cadherin[45], another critical component of cell junction, we examined the E-cadherin expression in E17.5 ovaries. As a result, a significantly reduced expression of E-cadherin was observed in hnRNPH1 cKO ovaries (Fig. 7i). Taken together, in combination with the previous finding that aberrant distribution of β-catenin and/or E-cadherin could cause the globally prenatal oocyte attrition[46,47], the data indicate that the abnormal splicing of genes functionally related to cell junction in hnRNPH1 cKO oocytes is driving the early oocyte development and meiosis failure in hnRNPH1 cKO female mice.

**hnRNPH1 recruits PTBP2 and SRSF3 to regulate mRNA alternative splicing.** To decipher the underlying molecular mechanisms of how hnRNPH1 regulates the RNA alternative splicing of its target gene, we performed RIP-sequencing (RIP-seq) for hnRNPH1 using purified WT germ cells from P28 testes. Annotation of the RIP-Seq data identified a total of 3817 transcripts that were significantly enriched in the hnRNPH1 immunopreci-pitants (cutoff: fold change >2, P value <0.05) (Supplementary Data 8), and most of the hnRNPH1 binding sites (~73.7%) resided in the introns of genes encoding proteins (Fig. 8a). RNA-binding motif analysis indicated that hnRNPH1 preferred G-rich tracts of RNA (Fig. 8b), which is consistent with previous studies of

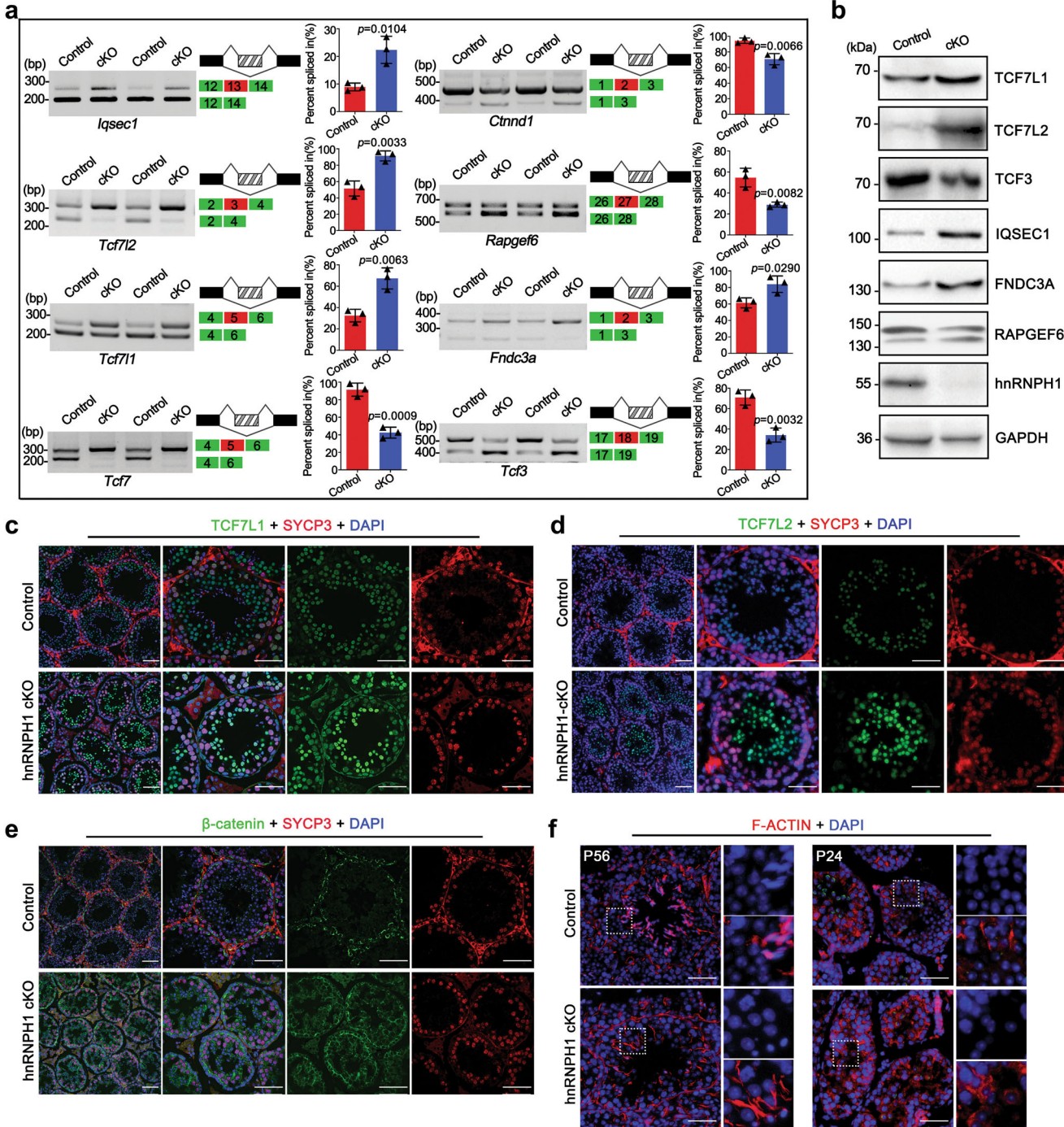

**Fig. 6 Ablation of hnRNPH1 in germ cells results in abnormal AS of cell junction-related genes and defective germ-Sertoli cell communications.**
**a** Representative examples of RT-PCR analyses for indicated AS events differentially regulated genes between control and hnRNPH1 cKO round spermatids are shown. Middle panels represent the schematic diagram of alternatively spliced exons detected by RNA-Seq analysis. Right panels show the quantification of percent spliced in (PSI). Data were presented as mean ± SD, $n = 3$ (three biological replicates). A two-sided Student's $t$-test was performed. **b** Western blotting analyses of the expression of TCF7L1, TCF7L2, TCF3, IQSEC1, FNDC3A, RAPGEF6, and hnRNPH1 in control and hnRNPH1 cKO round spermatids isolated from P25 mice. GAPDH was used as a loading control. **c–e** Co-immunofluorescence staining of SYCP3 with TCF7L1 (**c**), TCF7L2 (**d**), and β-catenin (**e**) on testis sections from control and hnRNPH1 cKO mice at P56. Scale bars = 50 μm. **f** Fluorescence microscopy to detect F-ACTIN (phalloidin) and DNA (DAPI) in testes from control and hnRNPH1 cKO mice at P56 (left panels) and P24 (right panels). Scale bars = 50 μm. Biologically independent mice ($n = 3$) were examined in three separate experiments (**b–f**).

hnRNPH1-RNA interactions[27]. Upon comparing RNA-seq and RIP-seq data, we found that ~23% (200 out of 881) and ~22% (205 out of 967) alternative splicing-changed genes were bound by hnRNPH1 in spermatocytes and spermatids, respectively (Fig. 8c). Notably, GO enrichment analyses revealed that these alternative

splicing-changed genes bound by hnRNPH1 were mainly related to RNA splicing regulation in both spermatocytes and spermatids (Supplementary Fig. 11a, b), indicating an essential role of hnRNPH1 in regulating many splicing factors, which may involve other splicing events. Moreover, a large number of differentially

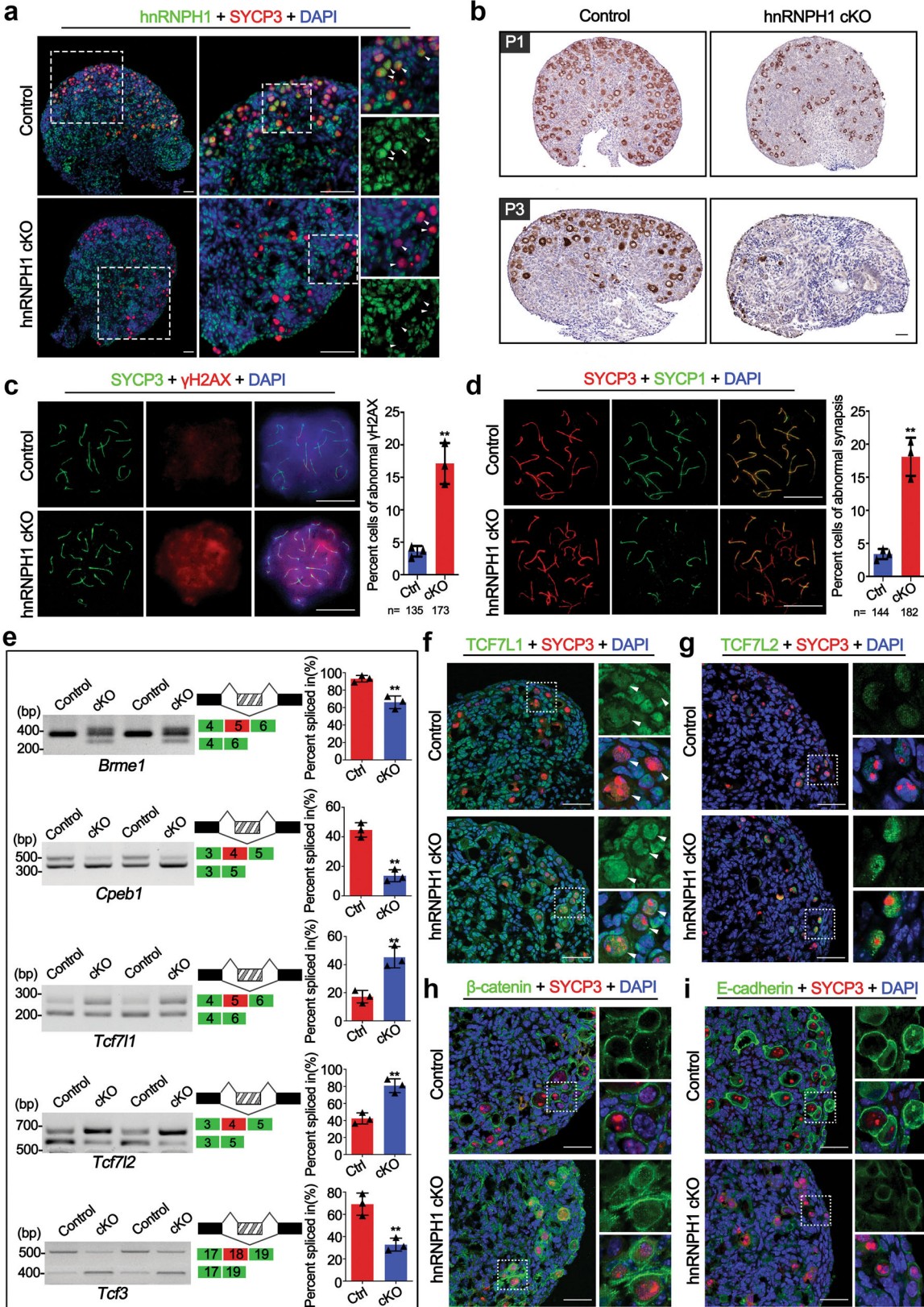

expressed transcripts were found to be bound by hnRNPH1 (Supplementary Fig. S11c), suggesting a splicing-independent mechanism underlying hnRNPH1 regulates gene expression.

Since PTPB2 has a similar role in RNA alternative splicing regulation in germ cells, PTBP2 was explored to see if it could bind hnRNPH1 target transcripts by comparing the reported CLIP-seq data of PTPB2[19] with the RIP-seq data. The results showed that many alternative splicing-changed genes were co-bound by hnRNPH1 and PTBP2 in spermatocytes and spermatids (Supplementary Fig. 11d). More interestingly, these genes were closely related to RNA splicing, meiotic nuclear division, and cell junction (Supplementary Fig. S11e, f). As most of the

**Fig. 7 hnRNPH1 is essential for the meiosis processes of early oocytes and female fertility. a** Co-immunofluorescence staining of hnRNPH1 and SYCP3 in ovaries from control and hnRNPH1 cKO mice at E17.5. Scale bars, 50 μm. Oocytes are indicated with white arrowheads. **b** Immunohistochemical staining of DDX4 in control and cKO ovaries at P1 and P3. Scale bars, 50 μm. **c** Co-immunofluorescence staining of SYCP3 with γH2AX in chromosome spread of pachytene oocytes from control and cKO mice at E17.5 is shown. The right histogram shows the quantification of the percentage of the cells with abnormal γH2AX signals. Scale bars, 5 μm. Data were presented as mean ± SD. A total of $n = 135$ control and $n = 173$ cKO pachytene oocytes were counted from three biologically independent mice for each genotype. A two-sided Student's $t$-test was performed, $**p = 0.002$. **d** Co-staining of SYCP3 with SYCP1 in chromosome spread of pachytene oocytes from control and cKO mice at E17.5 is shown. The right histogram shows the quantification of the percentage of the cells with abnormal synapsis. Scale bars, 5 μm. Data were presented as mean ± SD. A total of $n = 144$ control and $n = 182$ cKO pachytene oocytes were counted from three biologically independent mice for each genotype. A two-sided Student's $t$-test was performed, $**p = 0.001$. **e** RT-PCR analyses for indicated AS events differentially regulated genes between control and cKO ovaries at E17.5 are shown. Middle panels represent the schematic diagram of alternatively spliced exons. Right panels show the quantification of percent spliced in (PSI). Data were presented as mean ± SD, $n = 3$ biological replicates. A two-sided Student's $t$-test was performed, top to bottom: $**p = 0.0047$, $**p = 0.0010$, $**p = 0.0052$, $**p = 0.0030$, $**p = 0.0054$. **f–i** Co-immunofluorescence staining of SYCP3 with TCF7L1 (**f**), TCF7L2 (**g**), β-catenin (**h**), and E-cadherin (**i**) in ovaries from control and cKO mice at P0. Scale bars, 50 μm. Biologically independent mice ($n = 3$) were examined (**a**, **b**, **f–i**). Source data are provided as a source data file.

alternative splicing-changed genes that appeared in hnRNPH1 cKO germ cells were not directly bound by hnRNPH1 (Fig. 8c) but likely resulted from alterations in the splicing factors, we next examined whether some splicing factors were abnormal in their alternative splicing levels. The results showed that compared with control, among eight examined genes, *Hnrnpa2b1*, *Hnrnph3*, *Clk1*, and *Rbm5* showed significant splicing-changes in hnRNPH1 cKO germ cells (Supplementary Fig. S12a). Interestingly, we found three RBM5 target genes, *Anks3*, *Rangap1*, and *Cftr*[17], also exhibited abnormal alternative splicing in hnRNPH1 cKO spermatids (Supplementary Fig. S12b). RIP-qPCR was performed and confirmed that these three genes were bound by RBM5 but not hnRNPH1 (Supplementary Fig. S12c, d), indicating that some genes with abnormal alternative splicing events in hnRNPH1 cKO germ cells are not directly regulated by hnRNPH1 but by its target genes, particularly the splicing factors.

Considering the ability of hnRNPH1 to interact with splicing factors PTBP2 and SRSF3 in testes, we further explored whether the alternative splicing-changed genes in hnRNPH1 cKO germ cells were bound and regulated by PTBP2 and SRSF3. Therefore, we performed RIP-qPCR using the anti-hnRNPH1, PTBP2, and SRSF3 antibodies to immunoprecipitate the pre-mRNAs from purified germ cells, respectively. We chose 11 alternative splicings changed genes included in our RIP-seq data for further qPCR assay. The results showed that 6 out of selected genes (*Spo11*, *Cpeb1*, *Rapgef6*, *Rbm5*, *Hnrnpa2b1*, and *Clk1*) were simultaneously enriched in hnRNPH1, PTBP2, and SRSF3 immunoprecipitants (Fig. 8d). Thereafter, the overexpression and knockdown experiments were performed in HEK293T cells in vitro to investigate whether PTBP2 and SRSF3 could regulate the above six genes at the splicing level. Interestingly, we found that the alternative splicing of four genes (*Spo11*, *Cpeb1*, *Rapgef6*, and *Rbm5*) was changed in the same direction when hnRNPH1, PTBP2, and SRSF3 were either overexpressed or knocked down, respectively (Fig. 8e and Supplementary Fig. S13a). Significant co-localization of hnRNPH1 and PTBP2 at the indicated exons of the four target genes was confirmed by the RIP-seq and CLIP-seq experiments (Fig. 8f). These results revealed that hnRNPH1 target genes could be regulated by PTBP2 and SRSF3 at splicing levels. We further examined the binding affinity of the four target genes to PTBP2 and SRSF3 by RIP-qPCR assay in control and hnRNPH1 cKO germ cells. Unexpectedly, the enrichment level of all the four hnRNPH1 target genes bound to PTBP2 and SRSF3 protein significantly decreased in hnRNPH1 cKO germ cells compared with that in the control (Fig. 9a, b). A similar phenomenon was observed when hnRNPH1 was knocked down in HEK293T cells in vitro (Supplementary Fig. S13b, c). These data suggest that hnRNPH1 could recruit PTBP2 and SRSF3 to the corresponding positions of target genes and regulate their mRNA splicing.

As mentioned above, abnormal alternative splicing of some genes was found to occur in the hnRNPH1 cKO ovaries and that hnRNPH1 interacts with PTBP2 and SRSF3 in E17.5 ovaries (Supplementary Fig. 10d). Moreover, the male and female hnRNPH1 cKO mice had similar phenotypes, both of which exhibited meiotic defects and impaired cell adhesion function. RIP-qPCR assays were therefore performed to detect whether some transcripts associated with hnRNPH1 in male germ cells can be bound by hnRNPH1 in ovaries. The results showed that the mRNA of 6 out of the 11 genes could be directly immunoprecipitated by hnRNPH1 (Supplementary Fig. 14a). More interestingly, the transcripts of three out of these six genes were also bound by PTBP2 and SRSF3 in ovaries (Supplementary Fig. 14b). Moreover, we found that compared with the control, the mRNA enrichment of two important genes, *Cpeb1* and *Tcf7l1*, were lower in both the PTBP2- and SRSF3-immunoprecipitate (Supplementary Fig. 14c). These results demonstrate that hnRNPH1 can combine PTBP2 and SRSF3 to jointly regulate the alternative splicing of some target genes during oogenesis, which suggests the underlying mechanism of hnRNPH1 in the female germline may be akin to that in the male germline.

## Discussion

Spermatogenesis is a complex process dependent on the coordinated regulation of gene expression at transcriptional and post-transcriptional levels[48,49]. Many RNA-binding proteins (RBPs), particularly some critical splicing factors, have been involved in regulating spermatogenesis and played a role in different types of spermatogenic cells to ensure the orderly differentiation of spermatogenic cells because each spermatogenic developmental stage has its own unique alternative splicing program[50]. For example, BCAS2 could regulate meiosis initiation by promoting appropriate pre-mRNA splicing in undifferentiated spermatogonia[18], while MRG15, which was highly expressed at the onset of round spermatid differentiation, contributes to spermiogenesis by participating in mRNA splicing[15]. In the current study, we found that hnRNPH1 is mainly expressed in spermatocytes and round spermatids and involved in the alternative splicing regulation during the meiotic and post-meiotic stages. Increasing evidence indicated that many RBPs have direct roles in transcription, as exemplified by the elucidated function of particular splicing regulators in transcription, including SRSF2[51], HNRNPK[52], HNRNPL[53], and RBFOX2[54]. Like transcription factors (TFs), RBPs also showed an obvious preference for hotspots in the genome, particularly gene promoters frequently associated with transcriptional output[32].

It is noteworthy that among the hnRNPH1-interacting proteins identified in our IP-MS data, some TFs related to gene silencing, such as *Cirbp* and *Smarca5*, have been reported to be involved in spermatogenesis[55] and oogenesis[56]. Interestingly,

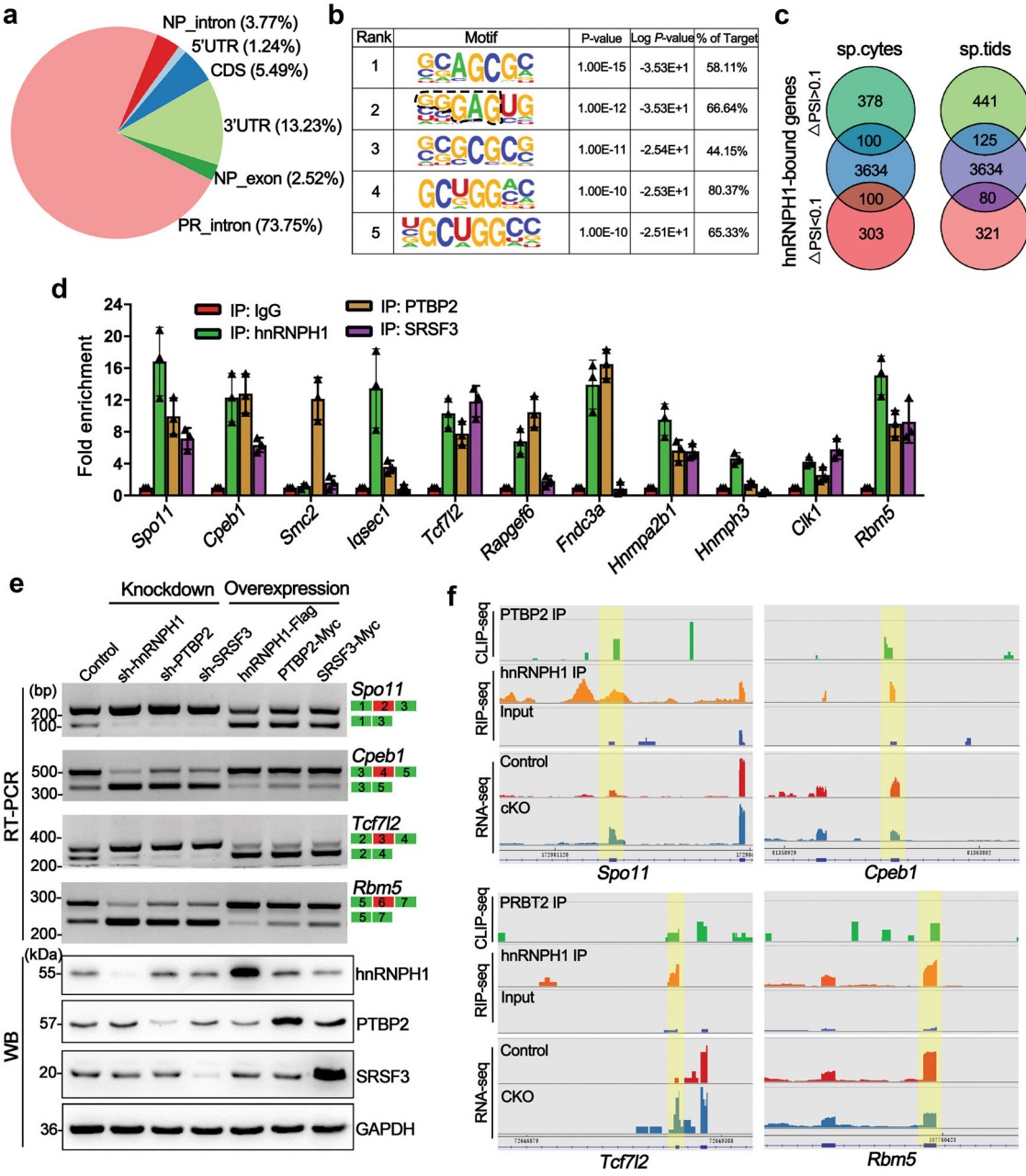

**Fig. 8 hnRNPH1 cooperates with PTBP2 and SRSF3 to regulate mRNA splicing. a** Pie-chat shows the summary of binding locations for hnRNPH1 RIP-sequencing (RIP-seq) in spermatogenic cells isolated from P28 WT testes. **b** Top five sequence motifs identified from RIP-seq peaks in spermatogenic cells are shown. **c** Venn diagrams showing overlap of hnRNPH1 bound genes and abnormal AS genes (with ΔPSI >0.1 and ΔPSI <0.1, respectively) in hnRNPH1 cKO spermatocytes (left) and spermatids (right). **d** Histograms show RIP-qPCR analyses of selected mRNA of 11 genes co-precipitated by anti-hnRNPH1, anti-PTBP2, anti-SRSF3 antibodies, and control IgG in RIP experiments performed from purified germ cells. Data were presented as mean ± SD, n = 3 (three biological replicates). **e** RT-PCR and WB analyses of splicing assays were performed in HEK293T cells transfected with the indicated minigenes and knockdown/overexpression-related vectors for hnRNPH1, PTBP2, and SRSF3. Biologically independent experiments (n = 3) were examined. **f** Alternative sites in four representative genes bound directly by hnRNPH1, SRSF3, and PTBP2 from RNA-, RIP-, and CLIP-seq using IGV software. Splicing sites are indicated by the yellow box.

most of the differentially expressed genes caused by hnRNPH1 deletion were upregulated; thus, it was speculated that hnRNPH1 might cooperate with specific TFs to mainly silence gene transcription in spermatogenic cells. Herein, the whole transcriptome profiling of spermatocytes and spermatids revealed a comprehensive splicing program that is susceptible to ablation of hnRNPH1 and affects genes of essential relevance for spermatogenesis. To support this, a large number of splicing events normally occurred during the transition from spermatocytes to

post-meiotic germ cells[14], some of which may be mediated by hnRNPH1 because of a spatiotemporal decreasing trend of hnRNPH1 expression from spermatocytes to round spermatids observed in this study. However, it is important to note that the RNA-seq data showed that hnRNPH1 ablation leads to dysregulation in the mRNA expression of a large number of genes, but most are likely not caused by abnormal alternative splicing, suggesting an independent role of hnRNPH1 in transcriptional regulation during spermatogenesis.

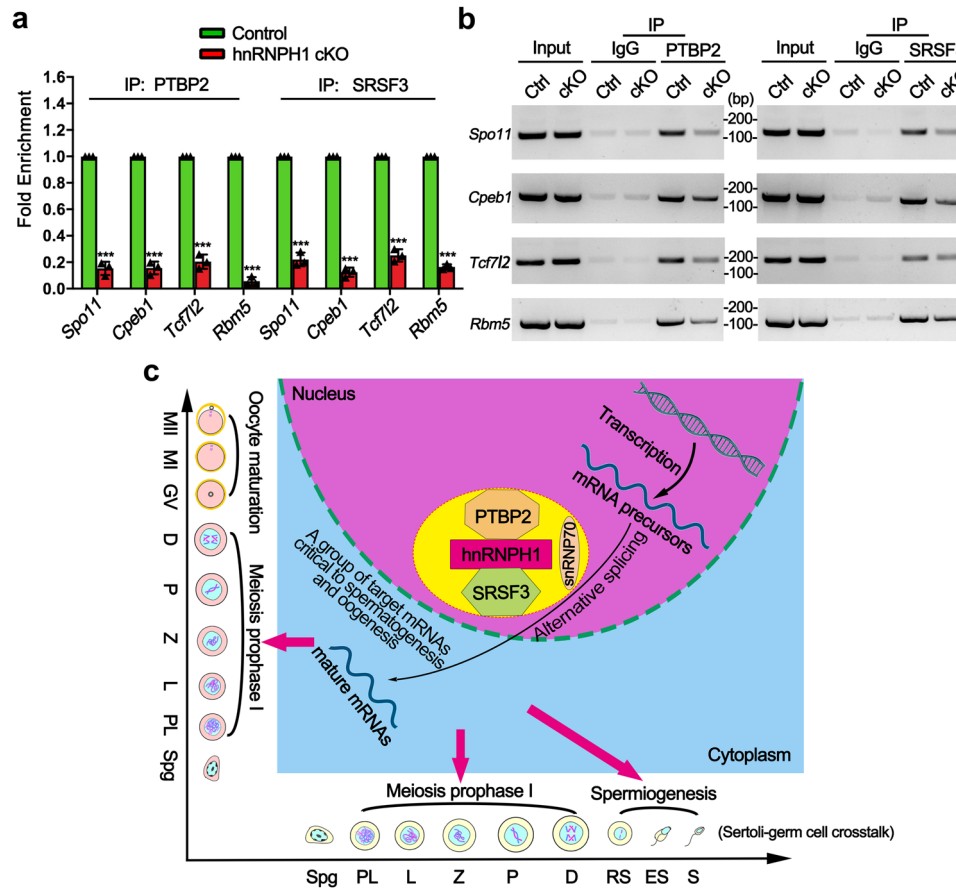

**Fig. 9 A proposed model of hnRNPH1 function in alternative splicing during spermatogenesis and oogenesis. a, b** RIP-qPCR (**a**) and RT-PCR (**b**) analyses of the association of the selected gene mRNAs with PTPB2 and SRSF3 in control and hnRNPH1 cKO testis are shown. Biologically independent mice ($n = 3$) were examined in three separate experiments. For (**a**), data were presented as mean ± SD. A two-sided Student's $t$-test was performed, left to right: ***$p = 8 \times 10^{-6}$, ***$p = 7 \times 10^{-6}$, ***$p = 1.4 \times 10^{-5}$, ***$p < 1 \times 10^{-6}$, ***$p = 1.2 \times 10^{-5}$, ***$p = 2 \times 10^{-6}$, ***$p = 1 \times 10^{-5}$, ***$p < 1 \times 10^{-6}$. **c** Schematic model showing that hnRNPH1 binds key splicing factors (e.g., PTBP2 and SRSF3) to coordinate proper alternative splicing of its target mRNA transcripts during spermatogenesis or oogenesis. Source data are provided as a source data file.

As an RNA-binding protein, hnRNPH1 can directly bind to the mRNAs of some target genes for regulation. Our RIP-seq data indicated that hnRNPH1 directly binds more than 3800 genes; however, only a small number of genes with differential expression or splicing-changes identified from hnRNPH1 cKO spermatogenic cells could bind to hnRNPH1. Of note, in the current study, hnRNPH1 was identified to bind and regulate the alternative splicing of genes encoding splicing factors (e.g., *Rbm5*, *Hnrnpa2b1*, *Clk1*, and *hnRNPH3*), suggesting hnRNPH1 may regulate the gene alternative splicing by cooperatively modulating some splicing factors themselves not directly regulate their target genes' splicing. It is reasonable to infer that dysregulation of these splicing factors inevitably would lead to more splicing errors in their target genes, thus amplifying the initial adverse effects and generating a vicious circle of aberrant splicing. As previously reported, hnRNPH1 is regarded as one of the core components of the splicing machinery regulating the dynamic balance of mRNA splicing, and in this study, it was found to play an equally critical role in spermatogenesis. In addition, while hnRNPH1 mostly binds intronic sequences, binding sites in coding and non-coding exonic regions were also observed in this study (Fig. 8a), raising the possibility that hnRNPH1 may have splicing-independent roles in posttranscriptional regulation in germ cells. Many potential interacting proteins related to translation were identified from the immunoprecipitants of hnRNPH1, which further supported this possibility. Moreover, a previous study has also clarified the role of hnRNPH1 in splicing, mRNA decay, and translation through a suite of high-throughput approaches[27]. Therefore, in addition to modulating alternative splicing, the possible functions of hnRNPH1 on the posttranscriptional regulation during spermatogenesis cannot be excluded. Of note, two key splicing factors, PTBP2 and SRSF3, were identified to interplay with hnRNPH1 in spermatogenic cells in this study, and many genes with identical splicing events regulated by hnRNPH1 and PTBP2 were found to be involved in several crucial signaling pathways related to reproduction. However, a number of splicing defects found in the hnRNPH1 cKO germ cells were not altered in the PTBP2 null germ cell, suggesting that hnRNPH1 and PTBP2 share some but not all splicing events. Another key splicing factor, SRSF3, has been reported to be bound by DDX5 to regulate splicing in spermatogonia[29] and was identified to regulate the splicing of hnRNPH1 target genes in the current study. More interestingly, the binding affinity of PTBP2 and SRSF3 to hnRNPH1 target genes (e.g., *Spo11*, *Cpeb1*, *Tcf7l2*, and *Rbm5*) was significantly reduced when hnRNPH1 was ablated (Fig. 9a, b), indicating that PTBP2 and SRSF3 could recruit the hnRNPH1 target genes to regulate its alternative splicing events. Nevertheless, PTBP2 and SRSF3 were only recruited to a small subset of hnRNPH1 target genes; the molecular mechanisms mediating the selective association of PTBP2 and SRSF3 with hnRNPH1 at specific target genes need to be elucidated in the future.

Of note, many genes showing abnormal alternative splicing identified in hnRNPH1 cKO spermatocytes were essential for the completion of meiosis. For example, Ehmt2[57], Spo16[58], Suv39h[59], and Brme1[43] have been reported to be involved in multiple meiotic progressions such as synapsis, recombination, and crossover formation. In this study, most of the genes that were verified to be splicing-changed had abnormal protein expression, which was the possible reason why some spermatocytes showed incomplete synapsis and DSB repair. Moreover, a group of hnRNPH1 cKO spermatocytes that appeared unsynapsis between X- and Y-chromosome was likely due to a lack of proper Spo11α subtype because aberrant Spo11 splicing could give rise to a higher risk of sex chromosomes aneuploidy[6]. Another interesting finding in the current study is that the female hnRNPH1 cKO mice were also infertile and showed similar meiotic defects as hnRNPH1 cKO male mice. Intriguingly, some of the meiosis-related genes (e.g., Brme1 and Cpeb1) also showed abnormal alternative splicing in the hnRNPH1 cKO ovaries, and both male and female mutant mice with these genes exhibited sterility[42,44]. These abnormal splicing events in meiotic genes resulting from hnRNPH1 ablation may be responsible for losing spermatocytes and oocytes.

During spermatogenesis, male germ cells must maintain stable attachment with Sertoli cells via testis-unique intermediate filament- and actin-based cell junctions to prevent sloughing of immature germ cells from the seminiferous epithelium[39]. In this study, a large number of prematurely sloughed round spermatids were observed in the epididymis of hnRNPH1 cKO testes, suggesting that hnRNPH1 may regulate the cell junction and adhesion between spermatogenic cells and Sertoli cells. Similarly, in the Ptbp2 mutant mice[16], spermatogenic cells showed increased apoptosis and premature release into the lumen, as observed in the hnRNPH1 cKO mice. Additionally, many genes functionally related to cell adhesion exhibited abnormal alternative splicing upon hnRNPH1 ablation in spermatogenic cells (particularly in round spermatids), further supporting the inference that the early spermatids released phenotype in hnRNPH1 cKO testes. Interestingly, whether in spermatocytes or round spermatids, many abnormal splicing events caused by hnRNPH1 ablation were consistent with that in Ptbp2 null testes. However, it is worth noting that many genes such as Tcf7l1 and Tcf7l2 were significantly upregulated in hnRNPH1 cKO spermatogenic cells, but this was likely further induced by the overexpression of β-catenin[34]. The germ cells sloughing from the seminiferous tubules accompanied by an overall increase of β-catenin in the epithelium resembled a common manifestation of disordered BTB structure. However, hnRNPH1, as a transcriptional or posttranscriptional regulatory factor, is unlikely to directly participate in the dynamic regulatory process of BTB opening and closing. Actin regulatory factors were thought to have critical roles in maintaining the adhesion of Sertoli cells with germ cells during their development and movement towards the lumen. Since most of the abnormally expressed proteins related to cell adhesion could regulate F-actin remodeling[60], it is plausible to observe disordered F-actin distribution in hnRNPH1 cKO testes similar to the features of Ptbp2 cKO testes showing a defect in germ-Sertoli cell communication attributed to the loss of Sertoli cell polarity. Another critical piece of evidence for this phenotype is that out of these abnormally spliced genes, Rapgef6[40], Fndc3a[41], and Rbm5[17] have been reported to mediate spermatid–Sertoli adhesion during mouse spermatogenesis and their expression was significantly changed in hnRNPH1 cKO germ cells due to abnormal splicing, which may have also contributed to the disorder of cell-cell junction and communication caused by hnRNPH1 ablation.

Moreover, although the abnormality of the first meiosis also occurred in the hnRNPH1 cKO female germ cells, the phenotype seems to be much more severe than that in the male germ cells because the early oocytes almost disappeared at P3. In hnRNPH1 cKO ovaries, the proportion of oocytes with abnormal localization of γH2AX or SYCP1 was only ~20%; thus, the defect of DSB repair or synapsis should not be the only cause of oogenesis arrest at an early stage. Surprisingly, many cell junction-related genes with aberrant alternative splicing in hnRNPH1 cKO testes, such as Tcf7l1, Tcf7l2, and Tcf3, were also abnormally spliced in hnRNPH1 cKO ovaries. This resulted in dysregulation of their target genes encoding β-catenin and E-cadherin, which have important roles in the maintenance between oocytes and granulosa cells. In addition, considering that aberrant distribution of β-catenin or E-cadherin could cause the globally prenatal oocyte attrition, the abnormal splicing of genes functionally related to cell junction was another critical reason for early oocyte loss in female hnRNPH1 cKO mice.

In summary, our study uncovers an essential role of hnRNPH1 in recruiting key splicing factors such as PTBP2 and SRSF3 to coordinate proper alternative splicing of its target mRNA transcripts, and establishes a model whereby hnRNPH1 controls a network of alternative splicing events in germ cells necessary to establish the correct protein isoforms required for meiosis and germ-Sertoli (or granulosa) cell crosstalk in mice (Fig. 9c). This research extensively expands our understanding of alternative splicing machinery regulating spermatogenesis through close cooperation between splicing factors.

## Methods

**Ethics statement**. All the experimental animal procedures were approved by the Institutional Animal Care and Use Committee of Tongji Medical College, Huazhong University of Science and Technology, and the mice were fed in specific pathogen-free facilities of Huazhong University of Science and Technology. All mouse experiments were conducted following the Guide for the Care and Use of Laboratory Animal guidelines.

**Mouse generation and genotyping**. Floxed hnRNPH1 mice (Cat# T001430) were purchased from the Model Animal Research Center of Nanjing University. In brief, the mice were generated by embryonic stem cells (ESCs) targeting and blastocyst injection. ESCs were targeted by carrying two loxP sites flanked in exon 6 and a neomycin selection cassette flanked by FRT sites in intron 5–6 of the Hnrnph1 gene. The Hnrnph1+/flox mice were obtained by chimera formation and germline transmission. Six-week-old mice were then crossed with FLP transgenic mice to remove the neomycin cassette and maintained on a C57BL/6 J background. Stra8-GFPCre mouse line in the C57BL/6 J background was obtained from Dr. Minghan Tong's Laboratory at the Center for Excellence in Molecular Cell Science, Chinese Academy of Sciences. Six-week-old Stra8-GFPCre males were first crossed with 6-week-old Hnrnph1flox/flox females to generate the Stra8-GFPCre; Hnrnph1+/flox males, then the 8-week-old Stra8-GFPCre; Hnrnph1+/flox male mice were bred with Hnrnph1flox/flox female mice to obtain the Stra8-GFPCre; Hnrnph1flox/△ (designated as hnRNPH1 cKO) male and female mice. For the global Hnrnph1 knockout mouse generation, Hnrnph1△/△ male mice were bred with Hnrnph1flox/△ female mice to generate Hnrnph1△/△ mice (designated as hnRNPH1 gKO). All mice were maintained at ambient room temperature (22 °C) with humidity of 40–70% and a light/dark cycle of 12 h/12 h. Mouse genotyping was performed by PCR amplification of genomic DNA extracted from mouse tails. The PCR primer sequences are shown in Supplementary Data 9.

**Fertility test**. Fertility was tested in control and hnRNPH1 cKO mice after 8 weeks old. Each hnRNPH1 cKO male or female mouse was caged with one WT C57BL/6 J female or male mouse (8–12 weeks), respectively, and the females were checked for vaginal plugs every morning. The plugged females were separated and single-caged, and the pregnancy was recorded. The fertility test lasted for at least 5 months.

**Histological analysis, immunostaining, and imaging**. Mouse testes, epididymides, and ovaries were collected immediately after euthanasia. Samples were fixed in Bouin's solution (Sigma, HT10132) at 4 °C overnight and then embedded in paraffin. Five-micrometer sections were prepared and mounted on glass slides. Hematoxylin and eosin (H&E) staining, periodic acid-Schiff (PAS) staining, and immunohistochemistry were performed using a standard protocol. For immunofluorescence staining, samples were fixed in 4% paraformaldehyde (PFA) in PBS overnight at 4 °C and were dehydrated sequentially and then embedded in Tissue-Tek optimal cutting temperature (OCT) compound. Cryosections were prepared at 5 μm. For antigen retrieval, slides were boiled in 0.01 M sodium citrate buffer

(pH = 6.0) for 20 min using a microwave. After a brief rinse with 1 × PBS, sections were blocked using a blocking solution (containing 3% normal goat serum and 3% fetal bovine serum in 1% bovine serum albumin) for 1 h at room temperature (RT). The sections were incubated with primary antibodies at 4 °C overnight. After rinsing with PBS, Alexa Fluor 488 goat anti-rabbit IgG (1:500; A32731, Invitrogen) and/or Alexa Fluor 594 goat anti-mouse IgG (1:500, A11032, Invitrogen) secondary antibodies were added and incubated for 1.5 h at RT, after rinsing with PBS and mounted using Vectorshield mounting media with DAPI (H1200, Vector laboratories). Laser confocal scanning images were captured using Leica Application suiteX (LAS X) with a digital camera (MSX2, Micro-shot Technology Limited, China). The details of primary antibodies used in this study are shown in Supplementary Data 10.

**TUNEL analyses.** TUNEL assay was performed with One Step TUNEL Apoptosis Assay Kit (Meilunbio, MA0223) following the manufacturer's instructions using testes and ovaries cryosections.

**Meiotic chromosome spread analyses.** Chromosome spreads were prepared as previously reported with slight modifications[61]. In brief, testicular seminiferous tubules were pretreated by hypotonic buffer (30 mM Tris, 50 mM sucrose, 17 mM trisodium citrate dihydrate, 5 mM EDTA, 0.5 mM DTT, and 0.5 mM phenylmethylsulphonyl fluoride (PMSF), pH = 8.2) for 60 min. The tubules were then cut into several short fragments and were suspended in 100 mM sucrose for dispersing to single cells and spreading to a thin cell layer on slides. Followed with 2% (w/v) paraformaldehyde solution containing 0.15% Triton X-100 treatment, the prepared slides were placed in a humidified chamber for 4 h at RT and washed with 0.4% Photo-Flo (Kodak, 1464510). The slides were then air-dried at RT for 15–30 min and then blocked with blocking solution (containing 5% normal donkey serum) for 1 h for immunostaining.

**Plasmids and minigenes construction.** Full-length and indicated fragments of *Hnrnph1* cDNA were cloned into a pCMV vector containing the N-terminal Flag epitope tag, and full-length and indicated fragments of *Srsf3* and *Ptbp2* cDNAs were cloned into a pCMV vector containing the N-terminal *c-Myc* epitope tag. HEK293T cells were transfected with indicated plasmids using Lipofectamine 2000 (Life Technologies). After 36 h, immunoprecipitation was performed using anti-Flag rabbit polyclonal antibody (5 μg antibody in 500 μl cell lysate, Cat#: 20543-1-AP, Proteintech), followed by a western blot to identify protein interactions. For minigenes construction, the *Spo11, Cpeb1, Tcf7l2, Rbm5, Anks3, Rangap1, Cftr,* and *Tcf3* minigenes were amplified from genomic DNA of adult mouse testis using primers shown in Supplementary Data 9. The minigenes were then cloned into pCDNA3.1(−) vector and validated by sequencing.

**Cell culture and transfection.** HEK293T cells were obtained from the Stem cell Bank of Chinese Academic Science (Cat# GNHu43), and cultured in the DMEM medium with 10% fetal bovine serum (Gibco, 10270106). Transfection was performed with Lipofectamine 2000 (Invitrogen) according to the manufacturer's instructions. For RNA interference, cells were transfected with appropriate siRNAs using Lipofectamine RNAiMAX (Invitrogen) and harvested 48 h later for analyses. The non-targeting siRNA (SIC001), targeting human hnRNPH1 siRNA (EMU072121), PTBP2 siRNA (EHU091671) and SRSF3 siRNA (EHU125071) were purchased from Sigma-Aldrich.

**Purification of spermatogenic cells and Sertoli cells.** The purification of spermatogenic cells was performed as described previously[62]. The STA-PUT method was used based on sedimentation velocity at unit gravity to purify the spermatogonia, spermatocytes, round spermatids, and elonged spermatids from WT adult mouse testes. The isolated spermatogenic cells with ≥90% purity were used for RT-qPCR analyses. In addition, pachytene spermatocytes and round spermatids from adult control and hnRNPH1 cKO mouse testes were purified by the STA-PUT method. Then the pachytene spermatocytes and round spermatids with ≥90% purity were used for RNA-seq analyses. The isolated spermatogenic cells were further confirmed by their distinct nuclear morphology (DAPI staining of nuclei).

Sertoli cell isolation was performed as described previously[63], with slight modification. In brief, the mouse testes at P21 were dissected and digested with collagenase IV (Sigma, Cat. No. C5138-100 mg) at 37 °C for 10 min. Then the seminiferous tubules were treated with trypsin (Sigma, Cat. No. 9002-07-7) and DNase I for 15 min at 37 °C in the shaker. After centrifuging and washing, the cell suspension was filtered and then cultured in 6 cm dishes at 35 °C in the DMEM/F12 medium containing 10% FBS. After a culture of 48 h, the cells were incubated with Tris-HCl (PH = 7.4) for 5 min to remove germ cells and other types of cells. Then the pure Sertoli cells were collected for the RT-qPCR experiment.

**RT-PCR and qPCR.** RNAs were extracted from indicated tissues using TRIzol reagent (Invitrogen), digested with Rnase-free DNase (Roche), and 1 ug of RNA was reverse-transcribed into cDNAs using the PrimeScript RT Reagent Kit (Takara) according to the manufacturer's protocol. RT-PCR primers designed to amplify two or multiple isoforms with different sizes are shown in Supplementary

Data 9. PCR products were quantified using Image J software. Splicing ratios were represented as PSI (Percent Spliced In) value, representing the percentage of a gene's mRNA transcripts that included a specific exon or splice site. qPCR was performed with SYBR green master mix (TaKaRa) on LightCycler@96 Real-Time PCR system (Roche) according to manufacturer's instructions.

**Western blotting.** The indicated samples were collected, and proteins were extracted by using RIPA buffer (CWBIO, Cat# CW2333S). Protein extracts were denatured with 5× SDS loading buffer (Beyotime, P0015L) at 100 °C for 10 min and run on a 10% SDS-PAGE, then transferred to PVDF membrane (Bio-Rad). After being blocked in 5% skimmed milk at RT for 1 h, the primary antibodies were incubated overnight at 4 °C. After washing, the membrane was incubated with the secondary antibody for 1 h at RT and then photographed using the Luminol/enhancer solution and Peroxide solution (ClarityTM Western ECL Substrate, Bio-Rad). The details of primary antibodies and secondary antibodies used in this study are shown in Supplementary Data 10.

**Immunoprecipitation.** Mouse tissues and cells were homogenized in cell lysis buffer, and the resultant cell extracts were treated with or without RNase A (1 μg/ml) at 4 °C for 1 h, followed by the lysates and were then clarified by centrifugation at 12,000×g. Thereafter, the relevant antibodies and pre-cleaned magnetic protein A/G beads were incubated overnight with the tissue lysates at 4 °C. The beads were washed with Cell lysis buffer for Western and IP (Beytime, P0013) with protease inhibitor cocktail (P1010, Beyotime) and then boiled in 5× SDS loading buffer (Beyotime, P0015L) for western blotting analyses.

**ChIP (chromatin immunoprecipitation).** Testes were dissected out of three P28 WT mice, and the tunica albuginea were removed. The seminiferous tubules were untangled and transferred into KREB buffer containing 0.5 mg/mL collagenase (Sigma) and incubated at 33 °C for 15 min followed by second digestion with 1 μg/mL DNase (Sigma) 0.5 mg/ml trypsin (Sigma) for 20 min. The single-cell suspension was then collected after the tubules were dispersed using a wide bore pipette, filtered through a 40 μm filter mesh cell strainer, and centrifuged at 1000×g for 15 min. Spermatogenic cells were roughly purified after the cell suspension was incubated with hypotonic shock solution (1 ml HBSS:10 Deionized water) for 1 min and centrifuged at 500×g for 3 min. For each ChIP experiment, $1 \times 10^5$ to $3 \times 10^5$ cells were used. To crosslink proteins to DNA, cells were fixed in 1% formaldehyde for 10 min at room temperature, then quenched by glycine for 5 min. Suspension cells were centrifuged at 1000×g for 5 min and washed two times with ice-cold PBS, then resuspended in buffer A plus protease inhibitor cocktail and DTT (Dithiothreitol) from SimpleChIP® Plus Enzymatic Chromatin IP Kit (Magnetic Beads). Nuclei were incubated on ice and pelleted by centrifugation at 1000×g for 5 min. After removing the supernatant, the pellets were resuspended in buffer B plus DTT and Micrococcal Nuclease and incubated for 20 min at 37 °C with frequent mixing to digest DNA to lengths of approximately 150–800 bp. Nuclei were pelleted by centrifugation at 15,000×g for 1 min and resuspended in ChIP Buffer and lysate sonicated with UH-100B Ultrasonic processor (Chincan) for 5 min (10 s "on" and 10 s "off") on ice to break the nuclear membrane. Immunoprecipitation was performed with Rabbit anti-hnRNPH1 antibody using the kit as described above. Immunoprecipitated and input DNAs were analyzed by real-time qPCR assay.

**CHIP-seq analysis.** For each sample, raw reads were processed to high-quality using Trimmomatic (http://www.usadellab.org/cms/index.php?page = trimmomatic) to remove adapters and perform trimming. Trimmed reads were mapped to the Human Genome Overview GRCm38 assembly (USCS mm10) (http://ftp.ensembl.org/pub/release-87/fasta/homo_sapiens/dna/) using STAR (https://github.com/alexdobin/STAR/releases) with default parameters and only uniquely aligned sequences were retained. DeepTools (http://deeptools.ie-freiburg.mpg.de/) was used for normalization to generate read density plot from BAM or bigwig files. For HNRNPH1 occupancy, ChIP-seq peaks were called by MACS2 (https://github.com/taoliu/MACS), with input used as the control. For MACS2, default parameters with a broad peak option and a broad cutoff of $P = 0.05$ were used.

**RNA immunoprecipitation, RIP-Seq, RNA-Seq, and bioinformatics analyses.** RNA immunoprecipitation (RIP) was performed using testes of 4-week-old mice. After the testes were dissected and decapsulated in 1 × PBS buffer at RT, the seminiferous tubules were lysed in buffer containing 100 mM KCl, 10 mM HEPES (pH = 7.0), 0.5% Triton X-100, 5 mM MgCl₂, 1 mM DTT, 0.5% NP-40, RNase inhibitor (100 U/ml) (Invitrogen), and EDTA-free proteinase inhibitor (Roche). Then, the testicular lysate was then passed through a 27.5-gauge needle five times to promote nuclear lysis, followed by brief sonication using a Bioruptor 200 (Diagenode). After incubation on ice for 20 min, the nuclear lysate was added with 50 ul beads for pre-cleared incubation at 4 °C for 1 h. For each reaction, 5 μg hnRNPH1 antibody (or IgG for controls) was incubated with protein G-Dyna beads in 1 ml NT2 buffer. After agitating at 4 °C for 4 h, control IgG and antibody-coated beads were incubated with testicular extracts and were agitated gently overnight at 4 °C. The next day, the bead complexes containing antibodies, target proteins, and RNA were washed for 20 min at 4 °C and repeated three times.

Protein-bound mRNAs were then extracted using an RNA extraction kit (ZYMO research). The purity and integrity of eluted RNA were assessed using an Agilent Bioanalyzer. One to two micrograms of cDNAs were synthesized from the extracted RNA using the Clontech SMARTer cDNA kit (Clontech Laboratories, CA USA, catalog# 634938), and adapters were removed through digestion with RsaI. According to the manufacturer's protocol, the cDNAs were fragmented using an ultrasonicator and profiled using an Agilent Bioanalyzer, followed by subjecting to Illumina library preparation using NEBNext reagents (New England Biolabs, Ipswich, MA USA, catalog# E6040). Thereafter, the size distribution, quantity, and quality of the Illumina libraries were measured by the Agilent Bioanalyzer. The libraries were submitted for Illumina HiSeq2000 sequencing (Otogenetics, Norcross, GA). Paired-end 100 nucleotide (nt) reads were generated, and the data quality was checked using FASTQC (Babraham Institute, Cambridge, UK). The raw data were then subjected to data analysis using Tophat2 and Cufflinks. Two biological replicates were analyzed for each sample. At least 40 ovaries were collected as a group for the RIP assay with E17.5 ovaries and were lysed in the lysis solution. Thereafter, a series of operational steps were carried out according to the treatment of spermatogenic cells.

For RNA-Seq, total RNA was extracted from pachytene spermatocytes and round spermatids that were sorted from adult control and hnRNPH1 cKO testes using TRIzol reagents (Invitrogen) following the manufacturer's protocol with three biological repeats. The concentrations and integrity of RNA were verified using a NanoDrop 2000 Spectrophotometer (Thermo Fisher Scientific). Thereafter, a total amount of 2 μg of RNA for each sample was used to prepare poly(A+)-enriched cDNA libraries using the NEBNext Ultra RNA Library Prep Kit for Illumina (New England Biolabs), and raw data (base pairs) were generated by the Illumina Hi-Seq 2500 platform. Raw reads were processed with cutadapt v1.9.1 to remove adapters and perform quality trimming, and trimmed reads were mapped to the UCSC mm10 assembly using HiSAT2 (V2.0.1) with default parameters. Differentially expressed genes for all pairwise comparisons were measured by DESeq2 (v1.10.1) with internal normalization of reads to account for library size and RNA composition bias. Differentially regulated genes in the DESeq2 analysis were defined as twofold changes, with an adjusted $P$ value of <0.05. Gene Ontology (GO) and KEGG analyses were performed using the database for DAVID. rMATS was used to analyze the alternative splicing events between control and hnRNPH1 cKO groups. To detect valid alternative splicing events, those with a false discovery rate (FDR) <0.05 and |△PSI| >10% were categorized as differential alternative splicing events, which were classified into five types: the retained intron (RI), skipped exon (SE), alternative 5′ splice site (A5SS), mutually exclusive exons (MXE), and alternative 3′ splice site (A3SS). The rmats2sashimiplot was used to convert the rMATS output into Sashimi plot.

**Statistical analyses**. The data were shown as the mean ± standard deviation (SD). GraphPad Prism 8.0 software (GraphPad, San Diego, CA, USA) was used for the statistical analyses. Significant differences between the two groups were analyzed using the two-sided Student's $t$-test and two-tailed Mann–Whitney $U$-test, and that between multiple groups was measured using one-way analysis of variance followed by Bonferroni post hoc tests. A value of $p < 0.05$ was considered statistically significant for any differences. $p$ values are denoted in figures or figure legends by $*p < 0.05$; $**p < 0.01$ and $***p < 0.001$.

**Reporting summary**. Further information on research design is available in the Nature Research Reporting Summary linked to this article.

## Data availability

All data needed to evaluate the conclusions in the paper are present in the article and/or the Supplementary Materials. All RNA-seq and RIP-seq data are deposited in the NCBI SRA (Sequence Read Achieve) database with the accession number PRJNA771927. Public ChIP-seq data sets used in this study are also accessible through the GEO Series accession number PRJNA491668. The String database [https://cn.string-db.org] was used for the analysis of protein–protein interaction networks. The source data are provided as a source data file with this paper. The authors declare that all data supporting the findings of this study are available within the article and its supplementary information files or from the corresponding author upon reasonable request. Source data are provided with this paper.

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

## Acknowledgements

We would like to thank Dr. Minghan Tong at the Shanghai Institute of Biochemistry and Cell Biology, Chinese Academy of Sciences, for kindly sharing the *Stra8-GFPCre* mouse line. This work, in part, was supported by grants from the National Natural Science Foundation of China (81971444 to S.Y., 31900606 to S.F., and 31801237 to X.W.), the Strategic Collaborative Research Program of the Ferring Institute of Reproductive Medicine, Ferring Pharmaceuticals and Chinese Academy of Sciences (FIRMSC200502 COV02 to S.Y.) and the Science Technology and Innovation Commission of Shenzhen Municipality (JCYJ20170818160910316 to S.Y.).

## Author contributions

S.F. and S.Y. conceived and designed the study. S.F., J.L., H.W., K.L., Y.G., Y.W., and X.W. performed all bench experiments and data analyses. S.F. and S.Y. wrote the manuscript. S.Y. supervised the project. All authors read and approved the manuscript.

## Competing interests

The authors declare no competing interests.
