## [Peer Review File · Nature Communications]

hnRNPH1 recruits PTBP2 and SRSF3 to modulate alternative splicing in germ cellsREVIEWER COMMENTS

Reviewer #1 (Remarks to the Author):

In this manuscript, Feng et al. characterize the role of RNA binding protein hnRNPH1 in spermatogenesis and oogenesis using mouse models. hnRNPH1 was found highly expressed in meiotic and post-meiotic cells in the testis and interacted through specific protein domains with various splicing factors including PTBP2 that have known roles in the germline. Body-wide *hnrnph1* deletion was embryonic lethal and conditional deletion in the male germline resulted in infertility due to a spermatogenic block. Transcriptional analysis suggested many splicing events were altered in *hnrnph1*-deficient spermatocytes and spermatids. Further, chromosome synapsis was perturbed during meiosis and germ cell-Sertoli cell adhesion was disrupted. RIP-Seq approaches were used to identify mRNA transcripts bound by hnRNPH1 and compared to those known to be bound by interacting splicing factor PTBP2.

Combined, this study represents a substantial body of work that provides detailed and novel insight into the roles of hnRNPH1 during spermatogenesis and identifies an essential role for hnRNPH1 in ovarian development. While multiple splicing regulators have been found to have essential roles during spermatogenesis, roles for hnRNPH1 are not described and this manuscript will be of interest to the field.

Points to be addressed:

1. Expression analysis of hnRNPH1 in the male germline in Figure 1a is rather unconvincing and requires further investigation. Negative controls for the antibody staining should be included in this dataset. It is stated that hnRNPH1 is absent from spermatogonia and elongated spermatids but available single cell RNA-Seq datasets of the mouse germline (e.g., Hermann et al., 2018), indicate that hnRNPH1 is broadly expressed during spermatogenesis from spermatogonia onwards. The authors should reassess this expression pattern based on these available datasets.
2. In Figure 3d, there are hnRNPH1-positive cells within the tubules of the conditional knockout mouse. What is the identity of these cells? Are all germ cells deficient for hnRNPH1? On a related note, why might the hnRNPH1 homolog hnRNPF be expected to compensate for hnRNPH1 (line 185?). Is there evidence from other systems for this?
3. Spermatocytes and round spermatids are isolated from control and *hnrnph1* deficient testes for transcription analysis. Data suggests altered expression of many genes and changes in splicing. Given that the *hnrnph1* knockout has a striking testis phenotype and disrupted testis architecture, are equivalent cell populations isolated from control and knockout animals for this analysis? Are the purity of these cell preparations the same? If slightly different populations/stages of spermatocytes and spermatids are isolated, the gene expression patterns would appear very different, but these changes would not be dependent on *hnrnph1* deletion.
4. In multiple places in the manuscript it is suggested that hnRNPH1 regulates/represses transcription of genes, e.g., lines 228-229 and 505-506, alongside regulation of splicing. However, no evidence is provided of this. Is hnRNPH1 recruited to the promoters of genes that are misregulated upon gene knockout? Are other splicing regulators known to perform similar transcriptional roles during spermatogenesis? With the data provided, these comments on transcriptional control are quite speculative but it is an important point as many genes are upregulated following hnRNPH1 deletion. Additional analysis of this transcriptional role should be provided.
5. hnRNPH1 is found to be required for synapsis of sex chromosomes but is apparently specifically excluded from the XY body. How are these points reconciled?

6. Based on RNA-Seq data, loss of hnRNPH1 does not affect splicing of Spo11, but analysis by RT-PCR found specific changes. This point is quite confusing as splicing changes should also be observable by RNA-Seq. This contradiction needs to be reconciled.

7. From analysis of the hnRNPH1 deficient ovary, it was concluded that changes in splicing of cell adhesion molecules underlies the failure in oocyte development. Given that many distinct genes are altered in the male germline following hnRNPH1 loss, it seems likely that other mechanisms are also relevant in the female germline. This point needs to be expanded and additional data provided. Alternatively, given that the analysis of the female germline is much less detailed than that of the male, the ovarian datasets could be removed and published separately elsewhere.

8. As a minor point, grammar and spelling should be checked carefully throughout the manuscript, and some language should be modified (e.g., "Inspiringly" and "enormously increased").

Reviewer #2 (Remarks to the Author):

The noteworthy results of this manuscript are that hnRNPH1 is essential for spermatogenesis and oogenesis, identification and validation of splicing targets (particularly Spo11), and co-regulation with PTBP2 and SRSF3.

The work presented is extremely detailed and thorough, and will be of interest to scientists investigating male infertility and splicing. The work presented does support the conclusions and claims. However, the abstract is a bit confusing in that it says hnRNPH1 binds to Spo11: this sounds a bit like a protein-protein interaction. If it is meant that hnRNPH1 binds to Spo11 mRNA then the Spo11 should be italicised, and it would help to add mRNA afterwards.

The methodology is sound, and meets accepted practice. There is enough detail for the methods to be reproduced.

Some of the text needs to be improved. There are some spelling mistakes that need to be corrected. Percent is spelled as persent in the figures, Figure 8 should say GAPDH, line 1029 should say FLAG not FALG, line 694 should say construction not costruction. The sentence including line 445 does not make easy reading, line 369 should say showing complete sterility.

RESPONSE TO REVIEWERS

GENERAL COMMENTS FOR ALL REVIEWERS

We thank both Reviewers for their careful consideration of our manuscript (NCOMMS-21-44427) and for their helpful comments. We also appreciate the Reviewer's overall enthusiasm and constructive criticisms, which have allowed us to improve the study significantly. Both Reviewers found our work to be interesting and novel to the field. However, Reviewer #1 has raised several important points regarding the interpretation of data for hnRNPH1 expression and functions. To address these critical points, we have reassessed its expression level in various spermatogenic cells in more detail and provided additional analysis of hnRNPH1's roles in transcriptional levels and female germline. We also have thoroughly edited the manuscript and revised several specific points in response to Reviewers # 1 and 2 as much as possible. We hope both Reviewers will find our revision to be satisfactory.

Our specific comments to each reviewer follow. Please note: Reviewer comments are in italics. Our responses are in blue font. The comments from the Reviewers have not been edited. We thank you again for your feedback and consideration.

Reviewer #1 (Remarks to the Author):

In this manuscript, Feng et al. characterize the role of RNA binding protein hnRNPH1 in spermatogenesis and oogenesis using mouse models. hnRNPH1 was found highly expressed in meiotic and post-meiotic cells in the testis and interacted through specific protein domains with various splicing factors including PTBP2 that have known roles in the germline. Body-wide hnrnph1 deletion was embryonic lethal and conditional deletion in the male germline resulted in infertility due to a spermatogenic block. Transcriptional analysis suggested many splicing events were altered in hnrnph1-deficient spermatocytes and spermatids. Further, chromosome synapsis was perturbed during meiosis and germ cell-Sertoli cell adhesion was disrupted. RIP-Seq approaches were used to identify mRNA transcripts bound by hnRNPH1 and compared to those known to be bound by interacting splicing factor PTBP2.

Combined, this study represents a substantial body of work that provides detailed and novel insight into the roles of hnRNPH1 during spermatogenesis and identifies an essential role for hnRNPH1 in ovarian development. While multiple splicing regulators have been found to have essential roles during spermatogenesis, roles for hnRNPH1 are not described and this manuscript will be of interest to the field.

Thank you so much for your appreciation of the significance of our work.

Points to be addressed:

1. Expression analysis of hnRNPH1 in the male germline in Figure 1a is rather unconvincing and requires further investigation. Negative controls for the antibody staining should be included in this dataset. It is stated that hnRNPH1 is absent from spermatogonia and elongated spermatids but available single cell RNA-Seq datasets of the mouse germline (e.g., Hermann et al., 2018), indicate that hnRNPH1 is broadly expressed during spermatogenesis from spermatogonia onwards. The authors should reassess this expression pattern based on these available datasets.

Thank you for pointing out this important aspect. To further clarify the expression pattern of hnRNPH1 during spermatogenesis, we reassessed its expression level in various spermatogenic cells in more detail. We first analyzed its mRNA expression profiles utilizing the available single-cell RNA-seq datasets from Dr. Hermann's group (Hermann et al, 2018). The results showed that hnRNPH1 mRNA exists in spermatogonia with a low level but is highly expressed in spermatocytes and mid spermatids and almost absent in late spermatids (**See the new Supplementary Fig.1f**). In addition, we isolated various types of testicular cells and analyzed the hnRNPH1 mRNA in those cell populations by qPCR (**See the new Supplementary Fig.1g**). The result is consistent with the single-cell RNA-seq data analyses.

To further explore whether hnRNPH1 protein is expressed in spermatogonia, we used the makers PLZF and c-KIT, which label undifferentiated- and differentiated-spermatogonia, respectively, to co-stain with hnRNPH1 in the P14 testis. The results showed that hnRNPH1 is weakly expressed in both types of spermatogonia (**See the new Supplementary Fig.2b**). Therefore, we conclude that hnRNPH1 mRNA is slightly expressed in spermatogonia, but the protein expression level is very low.

To verify the antibody specificity, we used two different sources of hnRNPH1 antibodies (anti-mouse and anti-rabbit) for immunofluorescence (IF) assay and analyzed the expression of hnRNPH1 protein in testis with a supply of their respective negative controls (omitting the primary antibody of hnRNPH1). Compared with the negative control, the signals of both hnRNPH1 antibodies are specific and show a similar distribution pattern (**See the new Supplementary Fig.2a**). In addition, the specificity of the two antibodies was verified in hnRNPH1 cKO testes, as shown in Fig.3d and the new Supplementary Fig.5a; hnRNPH1 was successfully knocked out in spermatogenic cells of the hnRNPH1 cKO mice. We included all the new data in this revision and modified the text accordingly (**See lines 124-131**).

2. In Figure 3d, there are hnRNPH1-positive cells within the tubules of the conditional knockout mouse. What is the identity of these cells? Are all germ cells deficient for hnRNPH1? On a related note, why might the hnRNPH1 homolog hnRNPF be expected to compensate for hnRNPH1 (line 185?). Is there evidence from other systems for this?

To identify the cell types of hnRNPH1-positive cells within the tubules of the hnRNPH1 cKO testes, we co-stained WT1 (a known Sertoli cells maker) and hnRNPH1 in the adult testes and found that these hnRNPH1-positive cells left in hnRNPH1 cKO testes were

Sertoli cells (WT1-positive) (**See the new Supplementary Fig.5a**). In addition, the new data also clearly showed that all germ cells are hnRNPH1 deficient in hnRNPH1 cKO mouse testes (**See the new Supplementary Fig.5a**).

Since hnRNPF is a member of the hnRNP F/H subfamily and shares high homology with hnRNPH1, and almost has the same functional domains as hnRNPH1. In many cases, hnRNPH seems to function together with the related hnRNPF (Fu & Ares, 2014). Previous studies revealed that hnRNPH and F are able to compensate for the lack of their counterpart (Nazim *et al*, 2017; Venables *et al*, 2008). Thus, we examined whether the expression of hnRNPF was changed in the hnRNPH1 cKO testes.

3. Spermatocytes and round spermatids are isolated from control and hnrnph1 deficient testes for transcription analysis. Data suggests altered expression of many genes and changes in splicing. Given that the hnrnph1 knockout has a striking testis phenotype and disrupted testis architecture, are equivalent cell populations isolated from control and knockout animals for this analysis? Are the purity of these cell preparations the same? If slightly different populations/stages of spermatocytes and spermatids are isolated, the gene expression patterns would appear very different, but these changes would not be dependent on hnrnph1 deletion.

In the current study, although a severe testicular phenotype appeared in the adult hnRNPH1 cKO mice, there are still some seemingly normal spermatocytes and round spermatids within the tubules in hnRNPH1 cKO testes (**Fig.3i**). In addition, STA-PUT is a well-known classical method for isolating all types of germ cells, and our lab had successfully isolated pachytene spermatocytes and round spermatids previously (Wang *et al*, 2021). Moreover, previous reports also used the STA-PUT to isolate the spermatocytes and round spermatids to study alternative splicing (Wang *et al*, 2020).

To check whether the germ cell populations we isolated from control and hnRNPH1 cKO testes for RNA-seq are equivalent, we performed the co-immunofluorescence staining of DDX4 (a germ cell marker) and SYCP3 (a spermatocyte marker) on the cell suspension isolated by STA-PUT. The result showed that the number of pachytene spermatocytes and round spermatids accounted for more than 90% of our isolated germ cells populations (**See the new Supplementary Fig.6a**), indicating that our RNA-seq data were reliable because of the purity of the germ cells we isolated was very high.

4. In multiple places in the manuscript it is suggested that hnRNPH1 regulates/represses transcription of genes, e.g., lines 228-229 and 505-506, alongside regulation of splicing. However, no evidence is provided of this. Is hnRNPH1 recruited to the promoters of genes that are misregulated upon gene knockout? Are other splicing regulators known to perform similar transcriptional roles during spermatogenesis? With the data provided, these comments on transcriptional control are quite speculative but it is an important point as many genes are upregulated following hnRNPH1 deletion. Additional analysis of this transcriptional role should be provided.

Thank you for raising these important comments. In fact, many RNA-binding proteins can also bind DNA and regulate the transcription of some genes. Our lab has recently also found that hnRNPU, another member of the hnRNPs family, could directly bind to the promoters of some genes to regulate their transcription(Wen *et al*, 2021). In this study, we found many differentially expressed genes in hnRNPH1 knockout spermatogenic cells, most of which were up-regulated. Thus, to explore whether hnRNPH1 binds DNA to regulate gene transcription, we re-analyzed the previously published ChIP-seq data of hnRNPH1 in K562 cells(Xiao *et al*, 2019). Unexpectedly, we found that hnRNPH1 has significant binding peaks near the promoter and the transcription start site (TSS) of more than 10,000 genes (**See the new Supplementary Fig.8a and Supplementary Table 7**), suggesting that hnRNPH1 can bind to the promoter of many genes. Combined with our RNA-seq analysis, we found that more than half of the up-regulated genes in both spermatocytes and round spermatids overlapped with the genes whose promoters are bound by hnRNPH1 (**See the new Supplementary Fig.8b-c**), suggesting that hnRNPH1 may directly bind to the promoters of these genes in germ cells. To test this, among these overlapping genes, we selected 10 up-regulated genes with the highest fold-changes for qPCR verification and found that the mRNA expression of 9 genes was indeed significantly increased in hnRNPH1 cKO germ cells (**See the new Supplementary Fig.8d**). We then performed ChIP-qPCR using isolated spermatogenic cells and found that the promoters of 4 of the tested genes can be bound by hnRNPH1 (**See the new Supplementary Fig.8e**), further suggesting that hnRNPH1 may bind the promoters of these genes in germ cells and regulate their transcription. We mentioned these additional analysis data in the main text (**See lines 286-301**). Because we did not find any pathways related to spermatogenesis in the GO term analysis of these differentially expressed genes (data not shown), we decided not to focus on the transcriptional regulation involved in hnRNPH1 in the current study. Of course, we cannot rule out some differentially expressed genes that have reported or unreported regulatory functions related to spermatogenesis. However, the transcriptional regulation mechanism of hnRNPH1 in spermatogenesis would be worthy of being further studied in another independent project in the future.

5. hnRNPH1 is found to be required for synapsis of sex chromosomes but is apparently specifically excluded from the XY body. How are these points reconciled?

Our IF results show that hnRNPH1 is specifically excluded from the XY body; thus, it is unlikely to regulate the synapsis of sex chromosomes directly. We propose that, among the target genes of hnRNPH1 that show abnormal expression or alternative splicing, some can regulate the synapsis of sex chromosomes. For example, abnormal splicing of *Spo11* has been reported to directly cause unsynapsis of sex chromosomes(Kauppi *et al*, 2011), and interestingly, *Spo11* was identified as a hnRNPH1 target gene showing abnormal splicing in hnRNPH1 cKO spermatocytes. Therefore, hnRNPH1 cannot directly bind to sex chromosomes to regulate their synapsis but indirectly affect this process through its target genes. We mentioned this point in the revision (**See lines 350-354**).

6. Based on RNA-Seq data, loss of hnRNPH1 does not affect splicing of Spo11, but analysis by RT-PCR found specific changes. This point is quite confusing as splicing changes should also be observable by RNA-Seq. This contradiction needs to be reconciled.

We appreciate that you have raised this critical point. RNA-seq has now been considered the standard method for analyzing the transcriptome gene expression quantification, but it still has some limitations, especially in data processing and analyses. Every RNA-sequencing analysis workflow will reveal a small but specific set of genes with inconsistent expression measurements. Some errors in RNA-seq quantification are probably caused by gene families that are often enriched for multi-mapped reads because the members in the same gene family have identical or close-identical sequences (Robert & Watson, 2015). In addition, various workflows showed about 15.1% to 19.4% of non-concordant genes that are usually smaller, have fewer exons, as well as lower expression levels (Everaert *et al*, 2017). Thus, in the current concept, the RNA-Seq data usually need to be verified by qPCR analysis to avoid the discrepancy derived from RNA-Seq quantification for some specific set of genes.

In our case, the finding that hnRNPH1 can regulate the alternative splicing of *Spo11* has been reported in detail (Cesari *et al*, 2020). Our RT-PCR result revealed a similar change of *Spo11* splicing in the hnRNPH1 cKO spermatocytes, although this change was not observed in our RNA-seq data. We, therefore, chose to believe the RT-PCR data because the RT-PCR is still considered a reliable method of choice for validating alternative splicing of candidate genes.

7. From analysis of the hnRNPH1 deficient ovary, it was concluded that changes in splicing of cell adhesion molecules underlies the failure in oocyte development. Given that many distinct genes are altered in the male germline following hnRNPH1 loss, it seems likely that other mechanisms are also relevant in the female germline. This point needs to be expanded and additional data provided. Alternatively, given that the analysis of the female germline is much less detailed than that of the male, the ovarian datasets could be removed and published separately elsewhere.

Thank you for your careful scrutiny of our work and your great suggestions. We agreed with the point that the relevant mechanism identified in the female germline is not as sufficient as in the male germline. Therefore, we expanded additional critical experiments to decipher the underlying mechanism of hnRNPH1 in female germlines following your suggestions in the revision. We performed immunoprecipitation for hnRNPH1 and found that, like in testes, hnRNPH1 could interact with PTBP2 and SRSF3 in E17.5 ovaries (**See the new Supplementary Fig.10d**). We also performed RIP-qPCR experiments to detect whether some transcripts associated with hnRNPH1 in male germ cells can be bound by hnRNPH1 in ovaries. The results showed that the mRNA of 6 out of the 11 genes could be directly immunoprecipitated by hnRNPH1 (**See the new Supplementary Fig.14a**). More interestingly, the transcripts of 3 out of these 6 genes were also bound by PTBP2 and SRSF3 in ovaries (**See the new Supplementary Fig.14b**). Furthermore, RIP-qPCR

experiments were carried out in control and hnRNPH1 cKO ovaries. The results showed that compared with control, the mRNA enrichment of two important genes, *Cpeb1* and *Tcf7l1*, were lower in both the PTBP2- and SRSF3- immunoprecipitate (**See the new Supplementary Fig.14c**). These additional data demonstrate that hnRNPH1 can combine PTBP2 and SRSF3 to jointly regulate the alternative splicing of some target genes during oogenesis, which suggests the underlying mechanism of hnRNPH1 in the female germline may be akin to that in the male germline. We added these parts of new data in the revision (**See lines 515-529**).

8. As a minor point, grammar and spelling should be checked carefully throughout the manuscript, and some language should be modified (e.g., “Inspiringly” and “enormously increased”).

The manuscript has been checked and polished by a professional editing service from the Editage company. All typos and grammars have been corrected.

Reviewer #2 (Remarks to the Author):

The noteworthy results of this manuscript are that hnRNPH1 is essential for spermatogenesis and oogenesis, identification and validation of splicing targets (particularly Spo11), and co-regulation with PTBP2 and SRSF3.

The work presented is extremely detailed and thorough, and will be of interest to scientists investigating male infertility and splicing. The work presented does support the conclusions and claims. However, the abstract is a bit confusing in that it says hnRNPH1 binds to Spo11: this sounds a bit like a protein-protein interaction. If it is meant that hnRNPH1 binds to Spo11 mRNA then the Spo11 should be italicised, and it would help to add mRNA afterwards.

Thank you so much for your careful evaluation of our work. We have corrected this sentence as suggested.

The methodology is sound, and meets accepted practice. There is enough detail for the methods to be reproduced.

Some of the text needs to be improved. There are some spelling mistakes that need to be corrected. Percent is spelled as persent in the figures, Figure 8 should say GAPDH, line 1029 should say FLAG not FALG, line 694 should say construction not costruction. The sentence including line 445 does not make easy reading, line 369 should say showing complete sterility.

We apologize for this carelessness. The revision has been carefully checked, and all typos and grammar have been corrected. Thanks again!

REFERENCES:

- Cesari E, Loiarro M, Naro C, Pieraccioli M, Farini D, Pellegrini L, Pagliarini V, Bielli P, Sette C (2020) Combinatorial control of Spo11 alternative splicing by modulation of RNA polymerase II dynamics and splicing factor recruitment during meiosis. *Cell Death Dis* 11: 240
- Everaert C, Luypaert M, Maag JLV, Cheng QX, Dinger ME, Hellemans J, Mestdagh P (2017) Benchmarking of RNA-sequencing analysis workflows using whole-transcriptome RT-qPCR expression data. *Sci Rep* 7: 1559
- Fu XD, Ares M, Jr. (2014) Context-dependent control of alternative splicing by RNA-binding proteins. *Nat Rev Genet* 15: 689-701
- Hermann BP, Cheng K, Singh A, Roa-De La Cruz L, Mutoji KN, Chen IC, Gildersleeve H, Lehle JD, Mayo M, Westernstroer B *et al* (2018) The Mammalian Spermatogenesis Single-Cell Transcriptome, from Spermatogonial Stem Cells to Spermatids. *Cell Rep* 25: 1650-1667 e1658
- Kauppi L, Barchi M, Baudat F, Romanienko PJ, Keeney S, Jasin M (2011) Distinct properties of the XY pseudoautosomal region crucial for male meiosis. *Science* 331: 916-920
- Nazim M, Masuda A, Rahman MA, Nasrin F, Takeda JI, Ohe K, Ohkawara B, Ito M, Ohno K (2017) Competitive regulation of alternative splicing and alternative polyadenylation by hnRNP H and CstF64 determines acetylcholinesterase isoforms. *Nucleic Acids Res* 45: 1455-1468
- Robert C, Watson M (2015) Errors in RNA-Seq quantification affect genes of relevance to human disease. *Genome Biol* 16: 177
- Venables JP, Koh CS, Froehlich U, Lapointe E, Couture S, Inkel L, Bramard A, Paquet ER, Watier V, Durand M *et al* (2008) Multiple and specific mRNA processing targets for the major human hnRNP proteins. *Mol Cell Biol* 28: 6033-6043
- Wang X, Li ZT, Yan Y, Lin P, Tang W, Hasler D, Meduri R, Li Y, Hua MM, Qi HT *et al* (2020) LARP7-Mediated U6 snRNA Modification Ensures Splicing Fidelity and Spermatogenesis in Mice. *Mol Cell* 77: 999-1013 e1016
- Wang X, Wen Y, Zhang J, Swanson G, Guo S, Cao C, Krawetz SA, Zhang Z, Yuan S (2021) MFN2 interacts with nuage-associated proteins and is essential for male germ cell development by controlling mRNA fate during spermatogenesis. *Development* 148
- Wen Y, Ma X, Wang X, Wang F, Dong J, Wu Y, Lv C, Liu K, Zhang Y, Zhang Z *et al* (2021) hnRNPU in Sertoli cells cooperates with WT1 and is essential for testicular development by modulating transcriptional factors Sox8/9. *Theranostics* 11: 10030-10046
- Xiao R, Chen JY, Liang Z, Luo D, Chen G, Lu ZJ, Chen Y, Zhou B, Li H, Du X *et al* (2019) Pervasive Chromatin-RNA Binding Protein Interactions Enable RNA-Based Regulation of Transcription. *Cell* 178: 107-121 e118

REVIEWERS' COMMENTS

Reviewer #1 (Remarks to the Author):

The authors have made substantial efforts to address all my comments with additional analyses. The manuscript has been significantly improved and I have no other major concerns.

Minor points:

1. The isolation procedure for spermatogonia and Sertoli cells for Supplementary Figure 1G is not detailed. In addition, in labelling of Supplementary Figure 1E, "Identity" is mis-spelled.
2. In line 288, the reference used as the source of hnRNPH1 ChIP-Seq data needs to be included.

POINT-BY-POINT RESPONSE TO REVIEWERS' COMMENTS

REVIEWERS' COMMENTS

Reviewer #1 (Remarks to the Author):

The authors have made substantial efforts to address all my comments with additional analyses. The manuscript has been significantly improved and I have no other major concerns.

Thank you so much for evaluation of our work again. We appreciate your suggestions to further improve our study.

Minor points:

1. The isolation procedure for spermatogonia and Sertoli cells for Supplementary Figure 1G is not detailed. In addition, in labelling of Supplementary Figure 1E, "Identity" is misspelled.

We added the details of the isolation procedure for spermatogonia and Sertoli cells into the "Methods" section (See lines 762-779). We also corrected the labeling errors for Supplementary Figure 1E.

2. In line 288, the reference used as the source of hnRNPH1 ChIP-Seq data needs to be included.

The reference was included in the revision, as suggested.